# Virtual fragment screening for DNA repair inhibitors in vast chemical space

Andreas Luttens [1,2,3], Duc Duy Vo[1], Emma R. Scaletti[4], Elisée Wiita[5], Ingrid Almlöf[5], Olov Wallner[5], Jonathan Davies[4], Sara Košenina [4], Liuzhen Meng[5], Maeve Long[5], Oliver Mortusewicz [5], Geoffrey Masuyer [4], Flavio Ballante [1], Maurice Michel[5], Evert Homan[5], Martin Scobie[5], Christina Kalderén[5], Ulrika Warpman Berglund[5], Andrii V. Tarnovskiy[6], Dmytro S. Radchenko [6], Yurii S. Moroz [6,7,8], Jan Kihlberg [9], Pål Stenmark [4], Thomas Helleday[5,10] & Jens Carlsson [1] ✉

Fragment-based screening can catalyze drug discovery by identifying novel scaffolds, but this approach is limited by the small chemical libraries studied by biophysical experiments and the challenging optimization process. To expand the explored chemical space, we employ structure-based docking to evaluate orders-of-magnitude larger libraries than those used in traditional fragment screening. We computationally dock a set of 14 million fragments to 8-oxoguanine DNA glycosylase (OGG1), a difficult drug target involved in cancer and inflammation, and evaluate 29 highly ranked compounds experimentally. Four of these bind to OGG1 and X-ray crystallography confirms the binding modes predicted by docking. Furthermore, we show how fragment elaboration using searches among billions of readily synthesizable compounds identifies submicromolar inhibitors with anti-inflammatory and anti-cancer effects in cells. Comparisons of virtual screening strategies to explore a chemical space of $10^{22}$ compounds illustrate that fragment-based design enables enumeration of all molecules relevant for inhibitor discovery. Virtual fragment screening is hence a highly efficient strategy for navigating the rapidly growing combinatorial libraries and can serve as a powerful tool to accelerate drug discovery efforts for challenging therapeutic targets.

Drug discovery efforts have generated millions of diverse compounds that are exploited in screening campaigns for novel therapeutic targets. However, high-throughput screening (HTS) of these libraries often yields low hit rates and weakly active compounds that require extensive optimization[1]. For these reasons, major focus has been put on development of techniques that can expand the accessible chemical space[2–4]. Advances in synthetic chemistry have enabled screens of increasingly larger chemical libraries and >30 billion compounds recently became available in make-on-demand catalogs[5]. These ultralarge compound collections are four orders of

[1]Science for Life Laboratory, Department of Cell and Molecular Biology, Uppsala University, BMC, Box 596, SE-751 24 Uppsala, Sweden. [2]Institute for Medical Engineering & Science and Department of Biological Engineering, Massachusetts Institute of Technology, Cambridge, MA 02139, USA. [3]Infectious Disease and Microbiome Program, Broad Institute of MIT and Harvard, Cambridge, MA 02142, USA. [4]Department of Biochemistry and Biophysics, Stockholm University, SE-106 91 Stockholm, Sweden. [5]Science for Life Laboratory, Department of Oncology-Pathology, Karolinska Institute, SE-171 77 Stockholm, Sweden. [6]Enamine Ltd., 02094 Kyiv, Ukraine. [7]Taras Shevchenko National University of Kyiv, Kyiv 01601, Ukraine. [8]Chemspace LLC, Kyiv 02094, Ukraine. [9]Department of Chemistry-BMC, Uppsala University, SE-751 23 Uppsala, Sweden. [10]Sheffield Cancer Centre, Department of Oncology and Metabolism, University of Sheffield, Sheffield, UK. ✉e-mail: jens.carlsson@icm.uu.se

magnitude larger than those tested by HTS, providing new opportunities for drug discovery.

Two central questions for drug discovery are how to efficiently find chemical probes in ultralarge libraries, and to what extent access to these collections improves chemical space coverage. Emerging techniques such as DNA-encoded libraries and large-scale virtual docking have identified potent leads from screens of several billion drug-like compounds[6–8]. However, considering that the total number of drug-like molecules has been estimated to range between $10^{23}$ and $10^{60}$, these libraries still only cover a very small fraction of chemical space[9,10]. The fact that the number of possible molecules grows exponentially with molecular size supports the use of fragment-based screening to initiate drug discovery. In fragment-based lead discovery, libraries of low-molecular-weight compounds (typically <250 Da or 9-16 heavy atoms) are first screened for ligands that bind to sub-pockets of the target protein. Fragment hits bind weakly to the protein, yet form high-quality interactions with the binding site. In the second step, the affinity and selectivity of the fragments are improved by increasing the size and complexity of the molecules[11–13]. As the estimated number of fragment-like compounds is only in the order of $10^{11}$, chemical space is more efficiently sampled in fragment screens than will ever be possible using drug-like collections[14,15].

Established techniques for fragment-screening, such as NMR, SPR, and X-ray crystallography are limited to collections of hundreds to thousands of molecules and restricted to physically available compounds[11]. The millions of fragments that are available in make-on-demand libraries are therefore inaccessible to traditional screening approaches. In contrast, structure-based virtual screening could rapidly evaluate large make-on-demand libraries and thereby reach further into chemical space. However, considering the small size and weak affinities of fragments, one potential caveat is that scoring functions may not have the accuracy needed to predict the affinities or binding modes of such compounds[16–19]. Fragment-based lead generation can also prove challenging and extensive chemical elaboration is generally required[12]. In this step, the >30 billion make-on-demand compounds could potentially be a valuable resource by providing access to readily available elaborations of fragments[20]. However, it is not clear if the number of fragment analogs and the diversity of these will be sufficient to enable successful fragment-to-lead optimization.

In this work, we explore the potential of vast chemical libraries using virtual fragment screening. Our approach is applied to discover inhibitors of 8-oxoguanine DNA glycosylase (OGG1), an enzyme that is part of the DNA damage response pathway. OGG1 recognizes the presence of the oxidized nucleobase 7,8-dihydro-8-oxoguanine (8-oxoG) in DNA and initiates repair by excision of the damaged base[21]. Recent studies demonstrate that inhibition of OGG1 is a promising strategy for the development of drugs against cancer and inflammation[22–24]. However, DNA-binding proteins are challenging drug targets due to their polar and flexible binding sites[25–27] and only a few OGG1 inhibitors have been identified to date[22,28–30]. We dock ultralarge compound libraries to the OGG1 active site and evaluate top-ranked compounds experimentally to identify starting points for inhibitor development. Crystal structures of OGG1-fragment complexes combined with docking of tailored chemical libraries enable rapid discovery of potent inhibitors displaying efficacy in cell models of cancer and inflammation. In addition, comparisons of different virtual screening strategies reveal efficient routes to identify chemical probes in vast chemical libraries.

## Results
### Ultralarge docking screens for OGG1 inhibitors
The determination of the crystal structure of mouse OGG1 in complex with a small molecule inhibitor (**TH5675**) enabled structure-based virtual screens for novel scaffolds (Fig. 1)[22]. At the time of the study, this was the only available crystal structure of OGG1 in complex with an inhibitor. A subsequently solved structure of inhibitor **TH5487** showed that the active sites of the mouse and human OGG1 were close to identical[23]. **TH5675** blocks the binding of the oxidized DNA substrates by occupying both the nucleobase- and furanose-binding regions of the active site. The active site is highly polar and adopts different shapes in the complexes with the inhibitor and DNA, which are properties characteristic of challenging drug targets (Fig. 1a, b)[26]. The molecular docking performance on the crystal structure was evaluated by redocking **TH5675** to the active site and assessing if the scoring function could enrich inhibitors over property-matched decoys[31]. The docking calculations were performed using DOCK3.7, which successfully reproduced the binding mode of **TH5675** and identified the inhibitor scaffold among decoys (Supplementary Fig. S1)[32].

Two ultralarge chemical libraries were docked to the OGG1 structure with the goal of identifying compounds binding to the nucleobase subpocket. The fragment-like (MW < 250 Da) library contained 14 million compounds and the lead-like library was composed of 235 million molecules that were larger in size and complexity (250 ≤ MW < 350 Da). The vast majority of these compounds have never been synthesized and were hence not available for testing by experimental screening methods. Each molecule in the two chemical libraries was represented by multiple conformations, which were scored in thousands of orientations in the active site. In total, 13 trillion fragments and 149 trillion lead-like complexes were evaluated by the docking scoring function. The top-ranked compounds from the screens were buried within the pocket occupied by the 4-iodo-phenylurea group of **TH5675**. Encouragingly, a large number of molecules containing similar motifs were among the top-ranked compounds. However, as such compounds were expected to bind to this pocket, we excluded all N-acylated six-membered arylamines from the library to bias the screen toward identification of novel chemotypes. For the fragment library, the 10000 top-ranked compounds (corresponding to 0.07% of the library) were clustered by topological similarity to identify a diverse set of candidate molecules. A set of 29 compounds was selected for experimental evaluation from the 500 top-ranked clusters based on visual inspection of the predicted complexes. Similarly, the 100000 top-ranked compounds from the lead-like library (corresponding to 0.05% of the library) were clustered and 36 compounds were selected from the 4000 top-ranked clusters. In the selection step, we inspected the complementarity of the compounds to the binding site and took into account contributions to ligand binding that are poorly described by the docking scoring function, in line with the standard practices of virtual screening[33]. Compounds were excluded from experimental evaluation based on widely-used criteria such as ligand strain, unsatisfied polar atoms in the binding site or the compound itself, and improbable tautomeric or ionization states[34]. In the selection of fragments, we focused primarily on compounds in the pocket occupied by the 4-iodo-phenylurea group of **TH5675**, which was the deepest cavity of the binding site and hence a promising anchoring point for an inhibitor. Whereas the fragments primarily occupied the deep cavity, the lead-like compounds were predicted to bind both in this pocket and other parts of the active site. The 65 selected compounds were available in make-on-demand catalogs and were successfully synthesized in 4–5 weeks (Supplementary Table S1). The compounds were first tested in a thermal shift assay based on differential scanning fluorimetry (DSF) at concentrations of 99 μM for lead-like compounds and 495 μM for fragments, which corresponded to conditions typically used in HTS and fragment screening campaigns, respectively[35]. Of these, compounds **1** and **2** from the fragment screen induced the largest shifts in thermal stability of OGG1 (2.8 and 1.6 K, respectively, Supplementary Table S2). In contrast, none of the lead-like molecules stabilized OGG1 significantly ($\Delta T_m$ <1 K). Based on these results, further characterization was focused on the hits from the fragment screen (Supplementary Table S1).

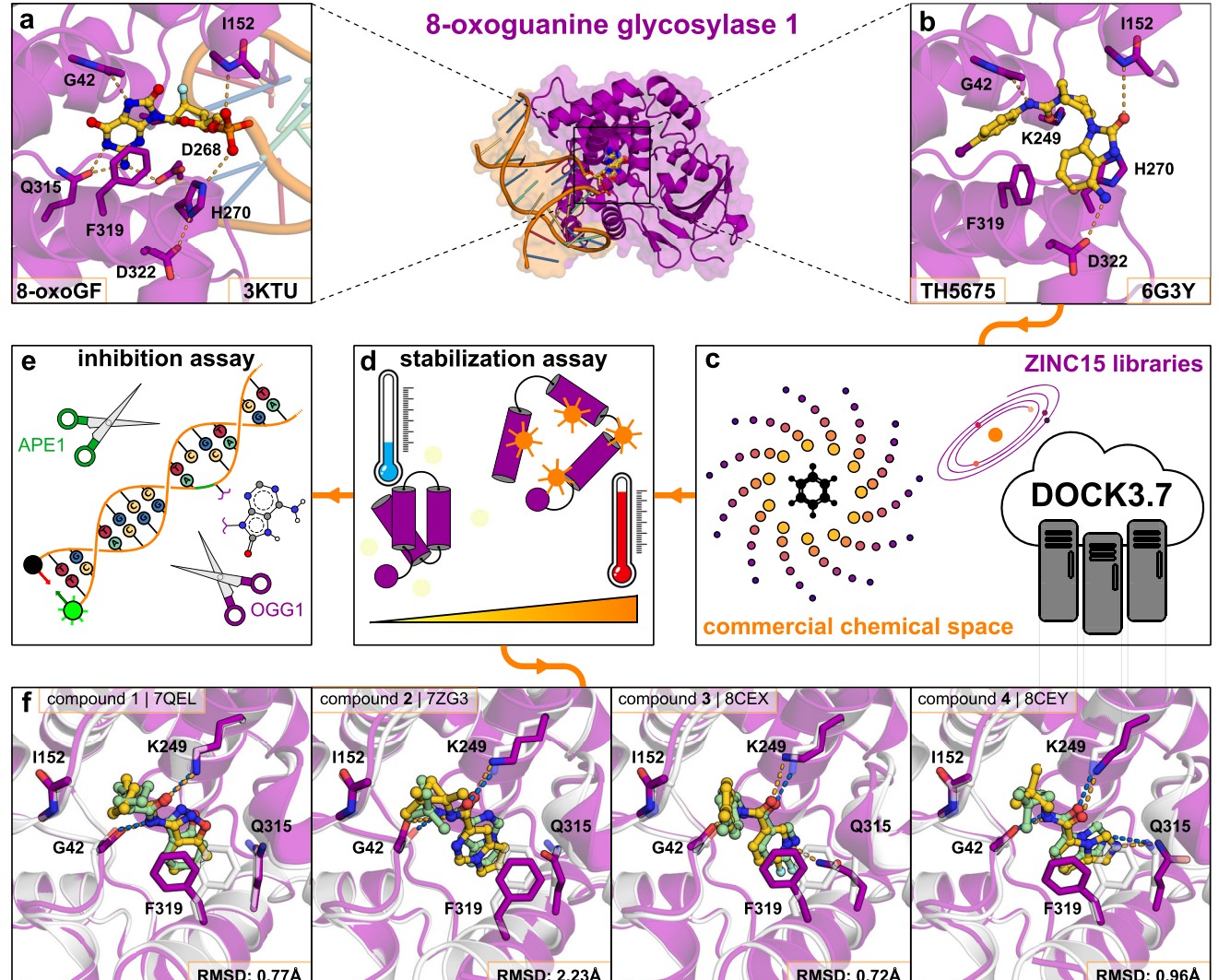

**Fig. 1 | Overview of virtual screening for OGG1 inhibitors. a** Structure of human OGG1 bound to DNA (PDB accession code: 3KTU). OGG1 is shown as a purple cartoon with key residues depicted as sticks. The oligonucleotide is shown in orange and the fluorinated 8-oxoguanosine is depicted as ball and sticks. Hydrogen bonds are shown as dashed lines. **b** Structure of mouse OGG1 in complex with inhibitor **TH5675** (PDB accession code: 6G3Y). **c** The ZINC15 libraries (14 million fragments or 235 million lead-like compounds) were docked to the OGG1 active site. **d** Compounds were evaluated in a thermal shift assay (differential scanning fluorimetry). **e** Compounds displaying thermal shift of OGG1 were evaluated in an enzyme inhibition assay. In this experiment, OGG1 cleaves 8-oxoadenine, followed by phosphodiester hydrolysis at the abasic site by the enzyme APE1, leading to disentanglement of the fluorophore-containing oligonucleotide. **f** Confirmation of the predicted binding modes by high-resolution crystal structures of mouse OGG1. The complexes predicted by docking (protein and fragments **1**–**4** are shown as white cartoons and green sticks, respectively) are depicted together with crystal structures (protein and fragments **1**–**4** are shown as purple cartoons and yellow sticks, respectively). Selected side chains are shown as sticks, and hydrogen bonds are shown as dashed lines. The accuracy of the predicted binding mode was quantified using the heavy atom root mean square deviation (RMSD) from the crystal pose.

## Crystal structures of fragments

Compounds **1, 2** and four additional fragments showing weaker stabilization of OGG1 ($\Delta T_m \geq 0.5$ K, Supplementary Table S1) were evaluated using protein crystallography, resulting in high-resolution structures of four complexes. Structures of compounds **1** (PDB code: 7QEL), **2** (PDB code: 7ZG3), **3** (PDB code: 8CEX), and **4** (PDB code: 8CEY) bound to mouse OGG1 were determined at resolutions ranging from 2.0 to 2.5 Å (Supplementary Table S3). For each fragment, an unambiguous binding mode could be determined from the electron density (Supplementary Fig. S2). The fragments represented four different chemotypes and formed similar interactions with the binding site. The compounds were anchored in the binding site by heterocyclic rings connected to an amide group, which formed hydrogen bonds to Gly42 and Lys249, and hydrophobic rings were positioned in the pocket at the entrance of the binding site. Two distinct binding site conformations were stabilized by the fragments. Whereas compounds **2** and **3** stabilized a binding site conformation that was similar to the structure used in the docking screen, alternative side chain rotamers for residues Phe319 and His270 were observed in the complexes with compounds **1** and **4**. The binding modes obtained by molecular docking agreed remarkably well with the crystallographic structures (Fig. 1f). Three predicted poses (compounds **1, 3**, and **4**) had root mean square deviations (RMSDs, ligand heavy atoms) to the crystal structure of less than 1 Å and all the key interactions with the binding site were captured in the fourth case (compound **2**, RMSD = 2.2 Å).

## Fragment elaboration by navigating in synthetically accessible chemical space

Our fragment growing approach was based on docking of computationally generated chemical libraries, which were either obtained from

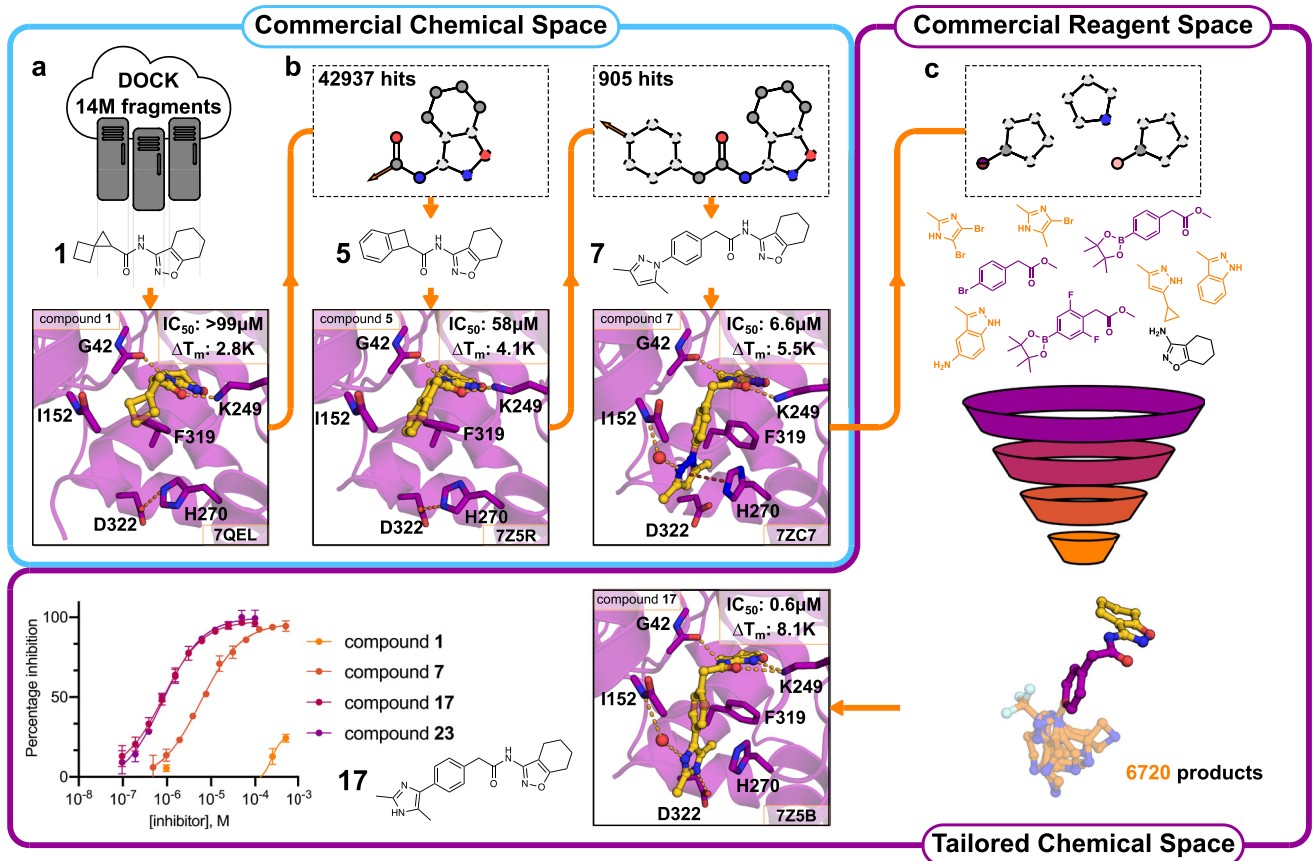

**Fig. 2 | Fragment expansion in synthetically accessible chemical space.**
**a** Docking of commercial chemical libraries containing 14 million (14 M) fragments led to identification of fragment **1**, a weak OGG1 inhibitor. **b** Crystal structures of mouse OGG1 bound to inhibitors enabled increasingly complex chemical pattern searches in commercial make-on-demand libraries. By stepwise increasing the size of the fragment, compounds **5** and **7** (IC$_{50}$ = 58 and 6.6 μM, respectively) were discovered and crystal structures of these were determined (PDB accession codes: 7Z5R and 7ZC7). **c** Suitable building blocks were retrieved using pattern matching based on the molecular topology of compound **7**. A tailored virtual chemical library

was constructed based on several coupling reactions, yielding 6720 products. Docking then guided selection of compounds from the virtual library, which were synthesized in-house and led to the discovery of compounds **17** and **23** (IC$_{50}$ = 600 nM, Table 1). A crystal structure of mouse OGG1 in complex with compound **17** (PDB accession code: 7Z5B) confirmed the computationally designed hydrogen bond to Asp322. Structure-guided elaboration leading to compounds **17** and **23** resulted in a >165-fold increase of inhibitory potency compared to fragment **1**. Source data are provided as a Source Data file.

searches among billions of make-on-demand compounds or via in silico reactions using commercial building blocks (Fig. 2). As fragment **1** induced the largest thermal shift in the DSF assay, we focused fragment expansion on this scaffold. Fragment **1** also inhibited OGG1 at high concentrations in an enzyme activity assay, but was too weak to determine an IC$_{50}$ value (Fig. 2a).

The potential for elaboration of fragment **1** was first explored by performing substructure searches in make-on-demand libraries for analogs of this compound. However, among the >11 billion available compounds (Enamine REAL database, version November 2019), there were only seven superstructures of fragment **1** and none of these could be docked successfully into the active site due to steric clashes. Based on analysis of the binding mode of fragment **1**, we further optimized the search to consider all compounds containing the N-(tetrahydrobenzisoxazole-3-yl)-formamide core, which anchored the compound in the active site. In this case, 42937 make-on-demand molecules matched the search pattern and were docked to OGG1. Elaborations that did not preserve the binding mode of the fragment core were automatically discarded and the remaining top-ranked complexes were inspected. Compounds were selected based on their complementarity with the active site and the potential for growing towards other subpockets. Five iterations of optimization were performed and a total of 62 make-on-demand compounds were experimentally evaluated (Table 1 and Supplementary Table S4).

Among the compounds from the first three iterations, **5** and **6** showed improved inhibitory potencies (58 and 36 μM, respectively). A crystal structure of compound **5** in complex with mouse OGG1 corroborated our binding mode predictions (PDB codes: 7Z5R, Fig. 2b), which positioned the core scaffold in the same pocket and orientation as observed for fragment **1**. Compounds **5** and **6** both contained a benzyl group that provided a starting point for extending toward other subpockets. Pattern searches in make-on-demand libraries for compounds containing the same topology as **5** and **6** identified 905 compounds. Of the 28 experimentally evaluated compounds from this set, compound **7** showed the largest improvement of potency with an IC$_{50}$ value of 6.6 μM and a thermal shift of 5.5 K at 99 μM in the DSF assay (Fig. 2b and Supplementary Table S5). A crystal structure of this inhibitor bound to mouse OGG1 (2.3 Å resolution, PDB code: 7ZC7) confirmed that the overall binding mode observed in the complexes with compounds **1** and **7** was maintained. Unexpectedly, the structure showed that the pyrazole ring of compound **7** extended into a subpocket formed by His270, Phe319, Leu323, and Asp322. The pyrazole formed pi-stacking interactions with His270, which disrupted its salt-bridge to Asp322, and this reorganization revealed opportunities for further potency optimization. However, searches for elaborations of compound **7** that could stabilize interactions in this pocket did not identify any suitable molecules in commercial chemical libraries. In total, a >15-fold

**Table 1 | Fragment elaborations, inhibitory potencies, and thermal stabilizations**

| Cmpd | Structure | pIC$_{50}$[a] | ΔT$_m$ (K)[b] |
|---|---|---|---|
| 5 | | 4.24 ± 0.18 | 4.1 ± 0.7 |
| 6 | | 4.44 ± 0.38 | 1.1 ± 0.5 |
| 7 | | 5.18 ± 0.05 | 5.5 ± 1.2 |
| 8 | | 5.19 ± 0.08 | 5.4 ± 0.5 |
| 11 | | 5.89 ± 0.06 | 4.8 ± 0.4 |
| 17 | | 6.22 ± 0.13 | 8.1 ± 0.1 |
| 23 | | 6.22 ± 0.10 | 9.0 ± 0.2 |

[a] pIC$_{50}$ values from enzyme inhibition assay.
[b] Thermal shift values from DSF experiments at a concentration of 99 μM. pIC$_{50}$ and ΔT$_m$ values are expressed as mean ± SD from at least three independent replicates.

improvement of inhibitory potency compared to fragment **1** was achieved by optimization using make-on-demand chemical libraries.

As commercial chemical space lacked elaborations of our most potent inhibitor, further optimization was driven by reaction-based enumeration of virtual libraries using commercial building blocks. Reagents compatible with Chan-Lam, Suzuki, Ullmann, or amide couplings were identified and reacted in silico to afford molecules with similar topologies as compound **7** (Supplementary Fig. S3). The resulting synthetically accessible virtual library contained 6720 molecules that were not available in make-on-demand catalogs (Fig. 2c). Visual inspection of the complexes predicted by docking guided the selection of building blocks required to synthesize the target molecules. As the docking was performed to the active site conformation of OGG1 observed in the complex with inhibitor **7**, several compounds were designed to extend into the additional subpocket identified in this structure. In total, 16 compounds (**8-23**, Supplementary Table S5) were successfully synthesized in-house and detailed synthetic procedures are provided in the Supplementary Information. Twelve compounds showed comparable or better potencies than compound **7** (Supplementary Table S5). Five of these were submicromolar

inhibitors (**17-20**, and **23**) and stabilized OGG1 by 7.4-9.0 K at 99 μM (Supplementary Table S5). Crystal structures of complexes were determined for two inhibitors (**8** and **17**). Compound **17** (IC$_{50}$ = 600 nM, ΔT$_m$ = 8.1 K at 99 μM) was designed to form an additional hydrogen bond to Asp322 by repositioning of a nitrogen atom in the five-membered aromatic ring and this interaction was confirmed by the crystal structure (2.4 Å resolution, PDB code: 7Z5B, Fig. 2c). The activity displayed by the most potent compounds (**17** and **23**, IC$_{50}$ = 600 nM, Fig. 2c) corresponded to a > 165-fold improvement of activity compared to fragment **1**. These compounds therefore exhibited inhibitor potencies comparable to **TH5487** (IC$_{50}$ = 340 nM, Fig. 3a). Notably, compounds **17** and **23** (ΔT$_m$ = 8.1 and 9.0 K, respectively) also showed greater thermal stabilization in the DSF assay than **TH5487** (ΔT$_m$ = 4.3 K), which may be due to the inhibitors representing different scaffolds[36]. Finally, compounds **17** and **23** exhibited physicochemical properties that were closer to the ideal profile of a lead compound, such as lower molecular weight and LogP, and higher lipophilicity ligand efficiency (Supplementary Table S6)[37–39].

## Selectivity and target engagement in cells
Selectivity for OGG1 was assessed by measuring the inhibition of four other DNA glycosylases and base excision repair enzymes (APE1, NEIL1, MTH1 and SMUG1, see Methods)[22,30]. Five inhibitors (**9, 11, 17, 18**, and **23**) were evaluated and three of these (**9, 11**, and **23**) did not display any significant inhibition of the four enzymes (IC$_{50}$ > 99 μM, Fig. 3b). For compound **23**, inhibitor interaction with OGG1 in cells was evaluated in two assays. Target engagement was confirmed using a cellular thermal shift assay (CETSA), which indicated a strong thermal stabilization of OGG1 by compound **23** in cells (Fig. 3c). Recruitment of OGG1-GFP to laser-induced DNA damage sites was impaired in U2OS cells treated with compound **23**, confirming an intracellular target engagement (Fig. 3d, e). In this assay, compound **23** showed activity comparable to the reference inhibitor **TH5487**.

## Therapeutic potential of OGG1 inhibitors
The anti-inflammatory and anti-cancer effects of the OGG1 inhibitors were evaluated in disease-relevant cell models. Inhibition of OGG1 limits binding to 8-oxoG in G-rich promoters of pro-inflammatory genes, which impedes the loading of the nuclear factor kappa-light-chain-enhancer of activated B cells (NF-κB) transcription factor. By blocking pro-inflammatory gene transcription, downstream inflammatory responses are attenuated[21]. The anti-inflammatory effect was evaluated in a TNF-α-induced NF-κB activation assay in HEK293T pGF NF-κB cells (Fig. 3f). All the tested compounds (**8-12, 14**, and **17-23**) showed dose-dependent inhibition of the inflammatory effect with EC$_{50}$ values ranging from 6.3 to 36 μM (Supplementary Table S5), and compound **23** was the most potent compound (EC$_{50}$ = 6.3 μM).

Cancer cells inherently have elevated levels of DNA damage, which can be attributed to disturbed redox homeostasis or oncogene-induced replication stress. For this reason, cancer cells are heavily reliant on functional DNA damage response and repair pathways[40]. We determined effects on the cell viability of three cancer cell lines (A2780, A549, and HCT116; representing ovarian, lung and colon cancer, respectively) and one non-transformed control cell line (BJhTERT, immortalized normal fibroblasts) for a subset of our OGG1 inhibitors (compounds **8-12, 14, 17-23**). Compound **11** induced the largest loss of viability in A2780 cancer cells (EC$_{50}$ = 5.4 μM) and was well tolerated (EC$_{50}$ > 70 μM) in non-transformed cells (Fig. 3g, h).

Compounds **17** and **23** were also evaluated for their in vitro pharmacokinetic properties, with the previously discovered inhibitor **TH5487** used as a reference (Supplementary Table S6). Compound **17** had slightly better cell permeability than **TH5487** and compound **23**. In human liver microsomes, compounds **17** and **TH5487** exhibited high metabolic stability (intrinsic clearance CL$_{int}$ = 23.9 and 4.6 μL/min/mg,

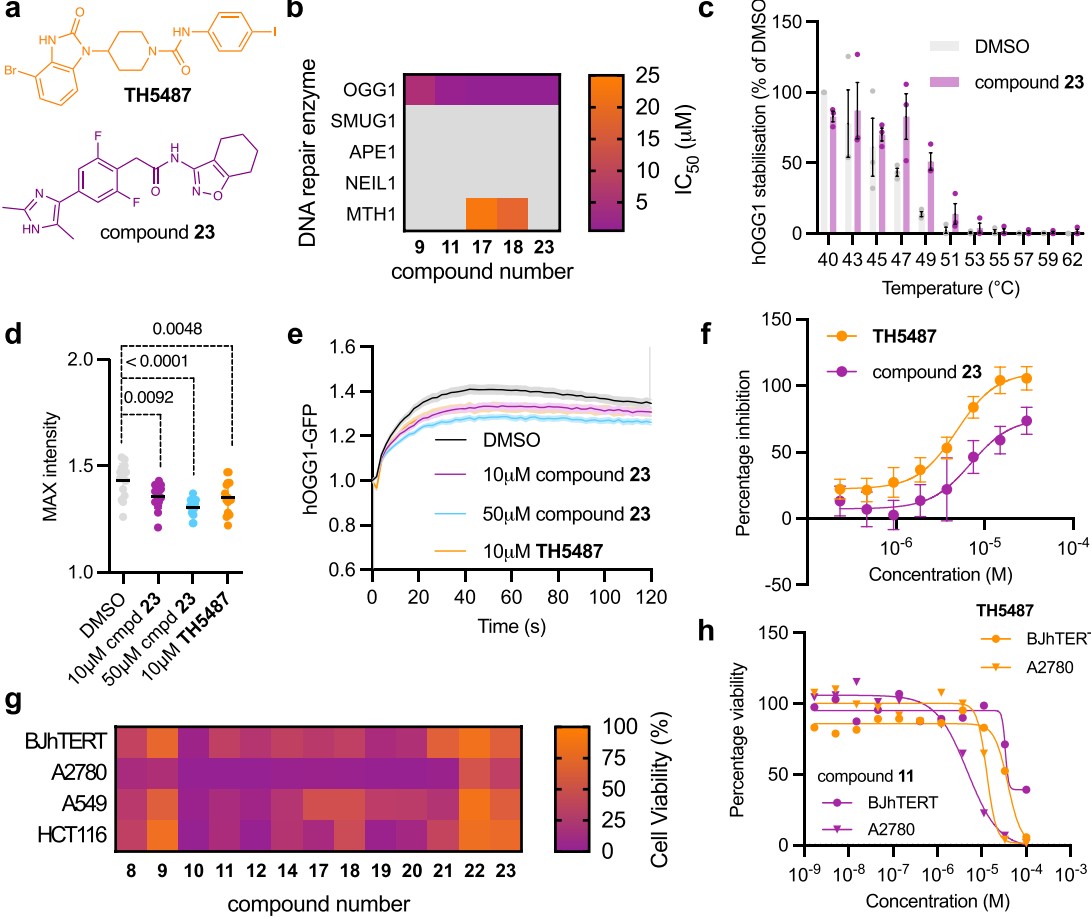

**Fig. 3 | Selectivity and cell activity of OGG1 inhibitors. a** Chemical structures of OGG1 inhibitors **TH5487** and compound **23**. **b** Inhibition of DNA repair enzymes (OGG1, SMUG1, APE1, NEIL1, MTH1, see Methods for abbreviations) by five OGG1 inhibitors. Mean $IC_{50}$ values from three independent experiments are shown. Grey boxes indicate $IC_{50} > 99\ \mu M$. **c** Shift in thermal stabilization of OGG1 in HL60 cells treated with OGG1 inhibitors. Mean values $\pm$ SEM from three independent experiments (dot plots) are shown. **d** Maximum fluorescence intensity of OGG1-GFP accumulation at laser-induced DNA damage sites after indicated treatments. Mean values of 15 cells for each condition from three independent experiments are shown. Statistical significance was determined using One-way Anova ($p = 0.0092$ for 10 $\mu$M compound **23**, $p < 0.0001$ for 50 $\mu$M compound **23**, and p = 0.0048 for **TH5487**). **e** Recruitment kinetics of OGG1-GFP to laser-induced DNA damage sites

in U2OS cells after 1 h pre-treatment with compound **23** and **TH5487**. Results of 15 cells for each condition from three independent experiments are shown $\pm$ SEM. **f** Anti-inflammatory effect of **TH5487** and compound **23** in NF-κB activation assay. Mean values $\pm$ SD from three independent experiments are shown. **g** The percentage cell viability of transformed and non-transformed cell lines upon treatment with different OGG1 inhibitors (100 $\mu$M). A2780, A549, and HCT116 represent ovarian, lung and colon cancer, respectively, and BJhTERT is a non-transformed control cell line. Mean values from two independent experiments are shown. **h** Cytotoxicity of **TH5487** and compound **11** (Table 1) in A2780 and BJhTERT cell lines. Mean values from two independent experiments are shown. Source data are provided as a Source Data file.

respectively) whereas compound **23** was moderately stable ($CL_{int}$ = 39.7 $\mu$L/min/mg). However, the low $CL_{int}$ value of **TH5487** is likely a consequence of the very high protein binding of this compound, as compared to **17** and **23**. Previous studies have identified high plasma protein binding and low solubility as a major limitation of **TH5487**[22]. These observations were confirmed in our ADME assays, which also showed that compounds **17** and **23** have considerably better properties. The thermodynamic solubility of compound **17** (667 $\mu$M) was >600-fold higher than for **TH5487** (<1 $\mu$M), which would be expected to precipitate at the highest concentrations used in biological experiments. In addition, **TH5487** had a very low unbound fraction in the plasma protein binding assay (fu = 0.1%), which was consistent with the reduction of inhibitory potency observed in the presence of serum albumin[23]. In contrast, **17** and **23** exhibited lower plasma protein binding levels (fu = 1.4 and 4.8%, respectively), *i.e.* the compound concentration available to inhibit OGG1 could be considerably higher in vivo compared to **TH5487** at the same dose.

## Comparison of strategies to explore chemical space

To assess the efficiency of different virtual screening strategies in more general terms, we explored paths in chemical space to identify our OGG1 inhibitors. The chemical space coverage of the fragment-based approach and virtual screens of multi-billion-scale libraries containing lead-like compounds were compared. As docking of several billion compounds is now feasible[41,42], we quantified what coverage of chemical space could be achieved with libraries of this size and assessed the impact of access to structural data during fragment elaboration.

We first analyzed the chemical space containing all molecules up to the size of the lead-like compound **17** (26 heavy atoms) and a fragment representing the core scaffold of the inhibitor (13 heavy atoms) (Fig. 4a). In order to estimate the size of each chemical space, we implemented an open-source version of the Generated Database (GDB) algorithm, which can enumerate all chemically stable molecules composed of H, C, N, O, S, and halogen atoms[14]. The algorithm was first used to generate all chemically stable molecules containing up to 11

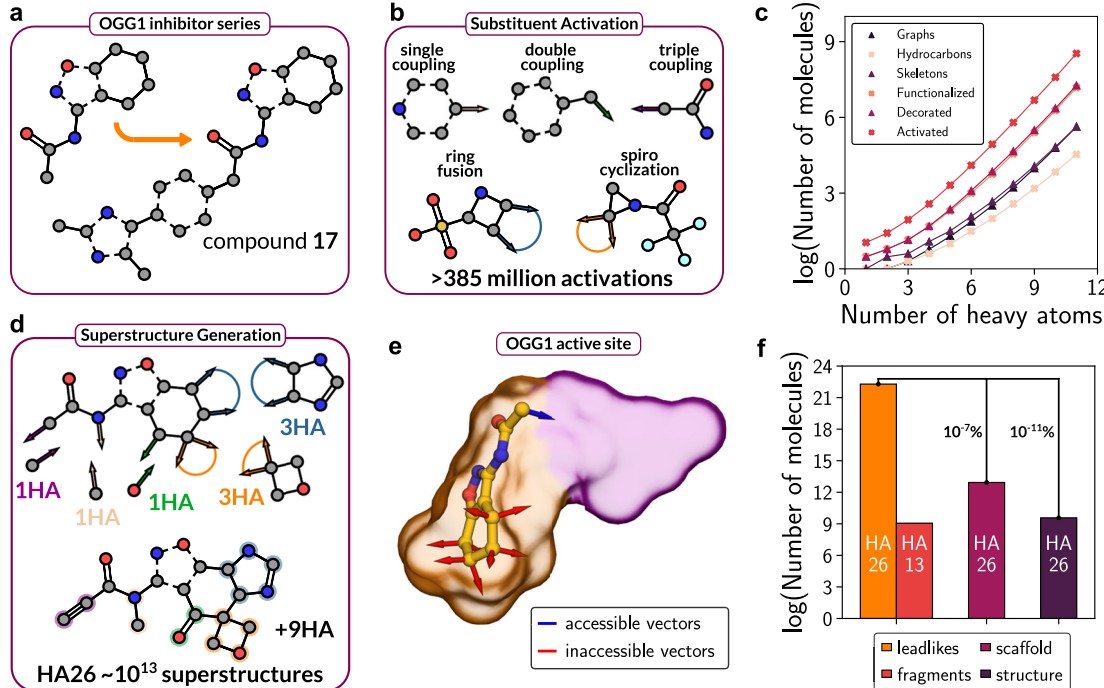

**Fig. 4 | Strategies to explore chemical space. a** Strategies to discover compound **17** based on screens of chemical libraries were analyzed. Screens of large chemical libraries with up to 26 heavy atoms (HAs) were compared to a fragment-based design approach using the tetrahydrobenzisoxazole core of compound **17** as a starting point. **b** Activated substituents can be attached to a core scaffold through five distinct mechanisms: introduction of a single (sand vectors), double (green vectors), or triple bond substituent (purple vectors), fusion of two ring fragments (blue vectors), and the spiro-cyclization of two ring fragments (orange vectors). **c** Generation and activation of all chemically stable molecules containing up to 11 heavy atoms. Curves represent the number of unique compounds in successive steps of molecule differentiation. **d** Generation of a superstructure by introducing substituents with a specified numbers of heavy atoms onto an activated scaffold. **e** Access to protein-ligand complexes enables exclusion of unsuitable (red) growing vectors due to steric hindrance (orange surface). Regions of the binding pocket accessible to suitable (blue) growing vectors are depicted as a purple surface. **f** Bar chart of estimated sizes of chemical spaces. The theoretical chemical spaces with up to 26 and 13 heavy atoms contain $10^{22}$ (orange) and $10^{9}$ (red) compounds. The core scaffold of compound **17** was estimated to have $10^{13}$ elaborations (raspberry), of which $10^{9}$ were compatible with the binding site (purple). Source data are provided as a Source Data file.

heavy atoms, and this library contained 21.2 million unique compounds, which is in agreement with previous estimates (Supplementary Table S7)[14]. Due to the combinatorial explosion of chemical space for larger molecules, the size of the libraries with up to 13 and 26 heavy atoms were calculated by extrapolation[15], resulting in estimates of $10^{9}$ and $10^{22}$ molecules for the fragment and lead-like space, respectively. A docking screen of one billion compounds could hence evaluate every possible fragment but is restricted to only $10^{-11}$% of the lead-like chemical space.

To further explore the fragment-based path to identify lead-like compound **17**, we developed the UniverseGenerator tool to generate all possible elaborations of a specific scaffold using our database of enumerated substituents. To construct a chemical space with a common-substructure, the library containing all possible molecules with up to 11 heavy atoms was prepared for connection onto a scaffold using activation-tags (Fig. 4b and Supplementary Fig. S4). Substituents could be attached through five distinct mechanisms: introduction of a single, double, or triple bond substituent, fusion of two ring structures, and the spiro-cyclization of two ring structures. In total, the 21.2 million substituents could be attached to a scaffold in more than 385 million distinct arrangements (Fig. 4c). Analogous to the activation of substituents, connection sites (*i.e.* growing vectors) on the fragment scaffold were first identified, followed by combinatorial enumerations to find all the 7154 unique configurations in which multiple substituents could be introduced (Supplementary Fig. S5). To finally construct all superstructures, an atom-distributing algorithm determined all possible combinations to attach differently sized substituents over a particular set of growing vectors (Fig. 4d and

Supplementary Fig. S6). We then explicitly generated compounds with up to 8 additional heavy atoms and estimated by extrapolation that the fragment has in the order of $10^{13}$ superstructures with up to 26 heavy atoms. To estimate the impact of access to structural information, we introduced elaboration constraints based on visual inspection of protein-fragment complexes determined for OGG1. Excluding growing vectors leading to steric clashes with the binding site (Fig. 4e) reduced the number of superstructures by four orders of magnitude, resulting in a database size ($10^{9}$ compounds) that is feasible to evaluate explicitly by molecular docking (Fig. 4f).

By combining the results of our analysis of the theoretical chemical space, we can compare the two virtual screening strategies. What are the odds of identifying our lead-like OGG1 inhibitor? By using a fragment-based approach, all theoretically possible fragment-like molecules up to 13 heavy atoms (one billion compounds) and every relevant elaboration of a single scaffold (an additional one billion compounds) could be evaluated by docking of two billion compounds in two steps. In contrast, a selection of two billion compounds from the lead-like chemical space ($10^{22}$) cannot be expected to contain any relevant representatives of this scaffold.

## Applicability of fragment-based virtual screening to other drug targets

We assessed if our approach could rapidly identify promising compounds of other drug targets. Three unrelated protein targets linked to cancer or inflammation (SMYD3, NUDT5, and PHIP) were selected[43–45]. Experimental fragment screening had been conducted for each target, and crystal structures of protein-fragment complexes were available.

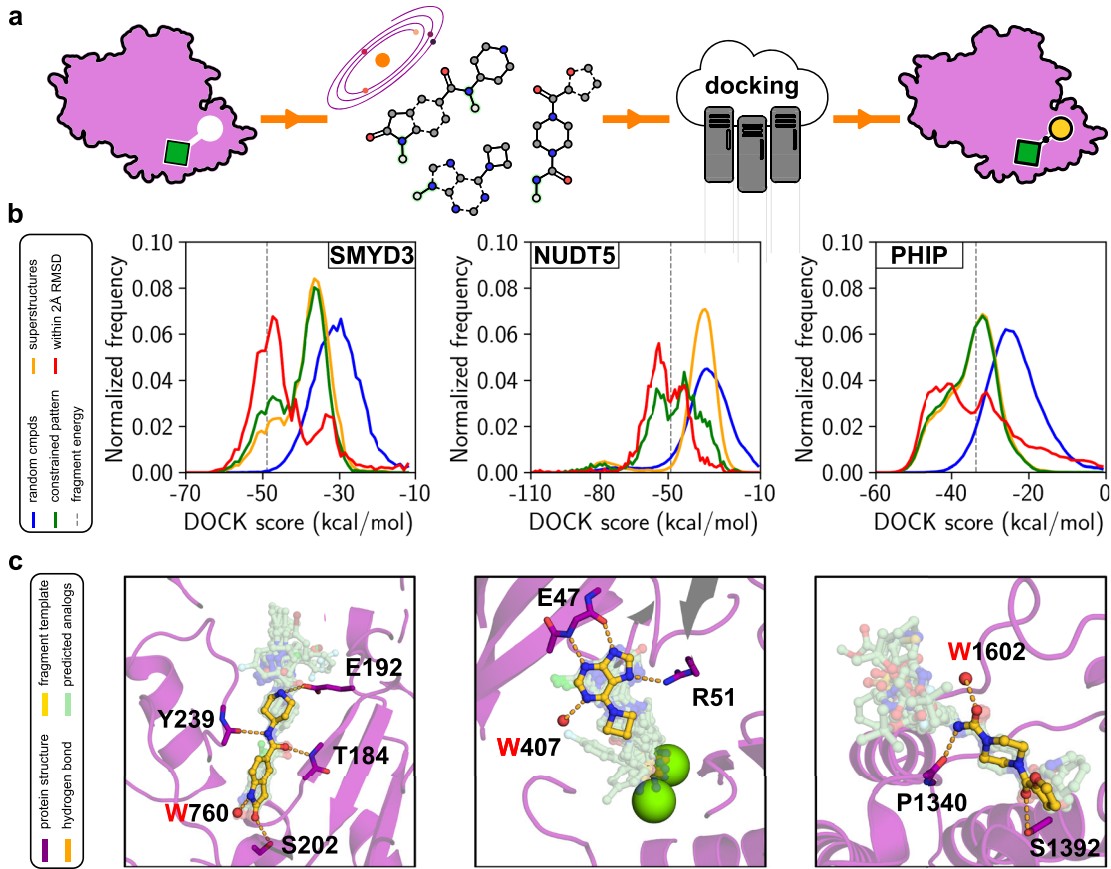

**Fig. 5 | Fragment-to-lead elaboration by docking of commercial chemical space. a** For each of the three targets (SMYD3, NUDT5, and PHIP), a SMARTS chemical pattern was generated based on the bound fragment, followed by searches in make-on-demand libraries containing billions of compounds for superstructures. The subset of matching compounds that was compatible with the accessible vectors was also identified (constrained superstructures). For example, moieties forming key hydrogens in the crystal structure of the protein-fragment complex (highlighted in green) were maintained in the constrained set. Molecular docking of the matching compounds enabled identification of suitable candidates for synthesis and experimental evaluation. **b** Normalized frequency distributions of molecular docking scores for SMYD3, NUDT5, and PHIP. Docking scores of the fragments are represented by dashed lines whereas scores of random molecules, superstructures, constrained superstructures, and superstructures that maintain the fragment binding mode are represented by blue, green, yellow and red curves, respectively. **c** Examples of predicted binding modes of elaborated fragments. The experimentally determined protein structures and bound fragments are shown as purple cartoons and yellow sticks, respectively. Key hydrogen bonds between the protein and fragments are indicated as yellow dashed lines. The predicted binding modes of top-scoring superstructures are shown as green sticks.

For each selected drug target, we evaluated if make-on-demand chemical libraries could enable efficient fragment elaboration (Fig. 5a).

The chemical structures of the co-crystallized fragments were used to perform substructure searches in the make-on-demand libraries, followed by docking of the matching compounds to the binding sites. Whereas only a small number of analogs was available in stock for each target (3-212 compounds), large sets of readily synthesizable fragment elaborations were identified by searches in make-on-demand libraries (9914-737407 compounds, Supplementary Table S8). Many of these had improved docking scores compared to the initial fragment (Fig. 5b) with maintained overall binding mode (814-45679 viable analogs, Supplementary Table S8). As in the case of OGG1, more efficient retrieval of suitable elaborations could be achieved by tailoring the search pattern through the exclusion of growing vectors incompatible with the binding site (Fig. 5b, c, Supplementary Table S8). For each target, compounds with both improved energy and maintained binding mode were retained while the number of compounds to dock was reduced, *e.g.* by 145-fold for NUDT5. For comparison, we also docked chemical libraries containing random molecules to each target, and substantially worse docking scores were typically obtained for these sets. The fragment-based approach had 6- to 125-fold higher enrichment of compounds with improved docking scores (Supplementary Table S8). Our approach therefore enables rapid identification of the most promising fragment elaborations for experimental evaluation.

## Discussion

Our efforts to identify enzyme inhibitors by structure-based virtual screening of vast chemical libraries resulted in three main observations. First, molecular docking of 14 million fragments identified four compounds binding to the OGG1 active site and crystal structures of complexes confirmed the computational predictions. Second, fragment elaboration was guided by docking calculations that made effective use of virtual libraries containing billions of readily synthesizable compounds. The combination of molecular docking and crystal structure determination led to selective submicromolar inhibitors with anti-inflammatory and anti-cancer activity. Finally, the results of the prospective virtual screens combined with analyses of the theoretical chemical space demonstrate the efficiency of a fragment-based docking approach.

The number of commercially available compounds continues to increase rapidly and the computational cost to perform virtual screens of these databases is becoming prohibitive. However, it should be noted that the fraction of the libraries that is most subject to inflation is

populated by drug-like molecules. In fact, 97% of the >30 billion compounds in the largest make-on-demand library have a molecular weight exceeding 350 Da. In contrast, there are only 50 million fragments in the database (MW < 250 Da), which are feasible to screen with available computational resources. Focusing on fragments will not only lead to substantially better chemical space coverage, but should also improve hit rates because drug-like molecules are less likely to bind due to their intrinsically higher molecular complexity[46]. In agreement with these ideas, we did not identify any OGG1 inhibitors with relevant activity from our docking screen of 235 million lead-like compounds. This library only contained two lead-like superstructures of the most potent fragment and none of these were top-ranked, illustrating that coverage of specific scaffolds is still limited in these libraries. The fragment docking campaign identified four diverse ligands and each of these represent starting points for development of unique inhibitor scaffolds.

Fragment-based drug discovery relies on efficient elaboration of hits, which can require synthesis of a large number of analogs[12]. The make-on-demand libraries enable rapid design-make-test cycles, but despite the billions of purchasable compounds, the availability of analogs to screening hits can still be scarce. As only a few elaborations of fragment **1** were commercially available, we used the predicted binding mode to identify the scaffold forming key interactions with the active site. Searches based on these molecular patterns enabled us to navigate to relevant regions of commercial chemical space. However, as the size and complexity of the optimized inhibitors increased, the number of relevant analogs in commercial space dwindled. At this point, chemical space coverage was further increased by generating tailored libraries of analogs made from accessible building blocks and this strategy proved efficient. Only 16 compounds were synthesized in-house to identify submicromolar OGG1 inhibitors, corresponding to a >165-fold increase of potency compared to fragment **1**.

The efficiency of our virtual screening approach is also illustrated by comparing to previous drug discovery efforts for OGG1 based on traditional approaches[22]. In the campaign by Visnes et al., two inhibitors representing a single scaffold were discovered from an HTS, corresponding to a hit rate of 0.01%. This was followed by extensive in-house medicinal chemistry efforts without any access to enzyme-inhibitor structures, leading to the discovery of the submicromolar inhibitor **TH5487**. In total, this drug discovery campaign involved the evaluation of >18,100 compounds experimentally. In comparison, we tested only 29 compounds from the fragment docking in experimental assays. As expected, the discovered fragments were less potent than the HTS hits, but represented four distinct scaffolds that each could serve as starting-points for optimization. Only 78 compounds, of which a majority originated from readily available make-on-demand libraries, were then synthesized to identify our submicromolar inhibitors. Our strategy therefore yielded a >1000-fold higher screening hit rate and required evaluation of >150-fold fewer compounds to identify inhibitors with potencies comparable to **TH5487**. Furthermore, our inhibitors also had more promising physicochemical and in vitro pharmacokinetic properties, such as considerably higher solubility and lower plasma protein binding. Together, these results highlight the advantages of our virtual screening technique and are consistent with the idea that a fragment-based approach results in lead compounds of higher quality compared to traditional approaches[35,37].

One of the central aims of this study was to compare strategies to screen the rapidly growing space of readily synthesizable molecules, which will surpass trillions in the near future. To address this question, we developed the UniverseGenerator, which enabled us to estimate the size of theoretical libraries containing all possible stable compounds. Out of the $10^{13}$ possible compounds representing our scaffold, we evaluated only a few thousand computationally and tested less than 100 experimentally, illustrating the limited coverage of commercial catalogs. Our chemical space analysis demonstrates that molecular docking enables evaluation of all theoretically possible fragments up to 13 heavy atoms, whereas virtual screens of libraries with billions of drug-like compounds (e.g. from make-on-demand catalogs) remain limited to sampling infinitesimal fractions of chemical space. By leveraging structural information, it is even possible to computationally evaluate all relevant elaborations of a fragment, which may become accessible for experimental testing as methods to synthesize complex molecules improve.

The use of structure-based virtual screening to identify and optimize fragments has been debated[16–19], and our study reveals strengths and weaknesses of this approach. In contrast to common belief, our results support the use of molecular docking as a tool to identify fragment ligands. Crystal structures revealed that the binding modes predicted by docking were strikingly accurate, and the hit rate was 5 to 10-fold higher than experimental fragment screening[27]. Similar observations were made in docking screens of fragment libraries for ligands of viral enzymes and G protein-coupled receptors[47–50]. Notably, our results from the fragment optimization step highlighted a well-known problem of molecular docking: the lack of binding site flexibility. Several distinct conformations of the active site were observed in crystal structures solved with different fragments. In two of these structures, the subpocket that our most potent inhibitor occupies was blocked due to changes in the side chain conformations. As the docking algorithm did not take protein flexibility into account, access to multiple structures of OGG1 representing different active site conformations was crucial for successful fragment elaboration. Fragment expansion should therefore consider several structures of the target if these are available or be performed using a flexible receptor algorithm[51].

The rapid growth of make-on-demand chemical libraries has stimulated the development of several novel virtual screening strategies[20,52–55]. For example, the V-SYNTHES technique uses an iterative approach to screen combinatorial compound libraries containing billions of compounds. Instead of screening all the compounds in the library, fragment-sized compounds representing the scaffolds available in the library are first docked to the binding site. In the second step, the top-scoring fragments are identified and libraries of larger molecules representing the same scaffold are docked to identify compounds for experimental evaluation. Our strategy and V-SYNTHES both employ a divide-and-conquer approach, which is a cornerstone of fragment-based drug discovery, and have complementary advantages. In the first step, both methods dock libraries of fragment-like compounds to the binding site. Whereas our approach uses commercially available libraries, V-SYNTHES is based on computationally generated fragments. A key benefit of our strategy is that false positives are identified experimentally at this stage, allowing us to focus on fragments with verified activity. In contrast, V-SYNTHES needs to rely on the accuracy of the docking scoring function. In the second step, V-SYNTHES depends on the predicted binding modes, whereas we iteratively solved structures of protein-ligand complexes. For OGG1, access to multiple experimental structures was essential because the challenging target binding site exhibited considerable induced-fit effects. Finally, V-SYNTHES is constrained to predict compounds that are available in the combinatorial library, which we found was a limitation in the optimization of the OGG1 inhibitors. The creation of tailored chemical libraries was required to obtain submicromolar OGG1 inhibitors, which has also been the case for other challenging targets[7]. Our strategy may be more suitable for difficult targets with flexible binding sites, but does require access to crystallography and sensitive experimental methods to detect weak binding. V-SYNTHES is preferable for targets with more well-defined binding sites and if structure determination is challenging, such as in the case of G protein-coupled receptors.

Access to chemical probes is essential for understanding biological systems and can enable drug discovery for novel therapeutic targets. The DNA repair enzyme OGG1 has been identified as an interesting drug target, but only a few inhibitors have been identified despite considerable efforts[22,28–30]. Here, we demonstrate how virtual fragment screening can rapidly identify inhibitors of challenging targets and achieve potencies sufficient to demonstrate activity in cell models. Our most advanced inhibitors are attractive leads for further development and can accelerate drug discovery efforts for this emerging target.

## Methods

### Ethics statement
Human plasma was collected from two voluntary healthy blood donors (non-smoking, citric acid) as part of routine procedures by Uppsala Academic Hospital. Written informed consent was obtained from both donors at time of collection. Human plasma samples were pooled before use in this study. As all human samples were anonymised, IRB approval at Uppsala University was not required.

### Molecular docking
A crystal structure (PDB accession code: 6G3Y, chain A) of OGG1 bound to an inhibitor (**TH5675**) was used in the docking screens[22]. Crystallographic waters and other solvent molecules were removed from the structure, except for the binding site water molecule with residue number 504. This water molecule was deeply buried and coordinated by several polar protein atoms, suggesting that displacing it would be challenging. Therefore, the water was treated as part of the binding site, enabling docked compounds to form hydrogen bonds with the oxygen atom. The atoms of **TH5675**, with the exception of the amino-benzimidazolone ring system, were used to generate 45 matching spheres in the deepest subpocket of the active site. DOCK3.7 uses a flexible ligand algorithm that superimposes rigid segments of a molecule's pre-calculated conformational ensemble on top of the matching spheres[32]. Histidine protonation states were assigned manually based on visual inspection of local hydrogen bonding networks. For example, His54 was protonated at the $N_\delta$ atom because of the hydrogen bonding interactions with the backbone carbonyl of Pro52 and amide proton of Leu16. Histidines 10, 13, 54, 97, 112, 179, 185, 195, 270, 276, and 282 were protonated at the $N_\delta$ atom, whereas histidines 119 and 237 were protonated at the $N_\varepsilon$ atom. The remainder of the enzyme structure was protonated by REDUCE[56] and assigned AMBER[57] united atom charges. The dipole moments of polar residues involved in recognition of **TH5675** were increased to favor interactions with these. This technique is common practice for users of DOCK3.7 to improve docking performance and has been used in previous virtual screens[33]. The partial atomic charges of the backbone amide of residue Gly42 were increased without changing the net charge of the residue. The atoms of the co-crystallized inhibitor were used to create a set of low protein dielectric spheres within 2 Å of the ligand and located near the protein surface to define the boundary between solute and solvent. Scoring grids were pre-calculated using QNIFFT[58] for Poisson-Boltzmann electrostatic energies, SOLVMAP[59] for ligand desolvation energies, and CHEMGRID[60] for AMBER van der Waals energies. Property-matched decoys of OGG1 ligands were generated using in-house scripts[31]. The obtained control sets were used to evaluate the performance of the docking grids by means of ligands-over-decoys enrichments. Enrichment values and predicted poses of ligands were used to select the final grid parameters.

DOCK3.7 was used to dock the fragment (MW ≤ 250 Da) and lead-like (250 Da <MW ≤ 350 Da) ready-to-dock subsets of ZINC15 to the OGG1 active site[61]. The libraries contained approximately 14 and 235 million commercially available molecules, respectively, and 14 and 212 of these were successfully docked. Fragments were screened using an orientational sampling parameter of 5000 matches and for lead-like compounds, this parameter was set to 1000 matches. For each fragment, 9344 orientations were evaluated on average, and for each of these orientations, an average of 202 conformations was sampled. Each lead-like molecule had on average 3694 orientations evaluated and, for each of these orientations, an average of 477 conformations was sampled. For each ligand, the best scoring pose was optimized using a simplex rigid-body minimizer. The virtual screens took 7066 (13 trillion fragment complexes) and 37355 (149 trillion lead-like complexes) core hours and could be performed in approximately 2 hours and 11 hours on 3500 cores, respectively. Top-scoring molecules were first filtered to eliminate compounds containing motifs present in the OpenEye toolkit's list of pan-assay interference compounds (PAINS), thereby reducing the risk of encountering false positives[62]. In addition, molecules containing an N-acylated six-membered arylamine were removed using SMARTS pattern matching to increase the novelty of top-scoring molecules. For each library, the remaining compounds were clustered based on Tanimoto similarity using Morgan2 fingerprints (1024 bits with a radius of 2), which were calculated using RDKit[63]. The clustering was performed using a toolkit from the DOCK3.7 software package[32]: The molecules are first ranked based on their docking scores. The best scoring molecule is the cluster head of the first cluster, and lower scoring molecules with a Tanimoto similarity greater than 0.5 to this molecule are included in this cluster. The first top-ranked molecule that is not part of this cluster becomes the cluster head of a new cluster and this procedure is repeated until all molecules have been assigned to a cluster. For the fragment screen, the top 10000 compounds were clustered and out of the 2577 resulting clusters, the 500 top-scoring cluster heads were visually inspected. For the lead-like library, we inspected the 4000 top-scoring cluster heads, originating from clustering 100000 top-ranked compounds into 20898 distinct clusters. In the fragment-to-lead generation step, crystal structures of OGG1 bound to fragment **1** (PDB accession code: 7QEL) and compound **7** (PDB accession code: 7ZC7) were prepared for molecular docking using the same protocols as described above. In these calculations, the dipole moment of the Ile152 backbone was increased to promote similar hydrogen bonding interactions as observed in the crystal structure of **TH5675**. For each docked fragment elaboration, the 50 lowest energy poses were retained. The lowest energy pose that had a common-heavy-atom RMSD value < 2 Å from the N-(tetrahydrobenzisoxazole-3-yl)-formamide core in fragment **1** was considered as the most relevant binding mode[7].

Crystal structures of the three protein targets (SMYD3, NUDT5, and PHIP) in complex with fragments were extracted from the PDB. Details regarding the preparation of crystal structures for molecular docking calculations are provided in Supplementary Table S9. Each elaboration of the bound fragment was docked using an orientational sampling parameter of 5000 matches and the 50 top-ranked poses were retained. In our analysis of the fragment elaborations, both the lowest energy pose and the lowest energy pose with a common-heavy-atom RMSD value < 2 Å to the bound fragment were considered as relevant binding modes.

### Cheminformatics and preparation of chemical libraries
In searches for elaborations of fragment **1**, chemical SMARTS patterns were constructed using structural information and an interactive SMARTS visualizer (https://smarts.plus)[64]. Chemical pattern matching was performed with OpenEye's OEToolkits and Enamine REAL space[65], enumerated (version November 2019). Matching molecules were filtered by using a PAINS-filter from OpenEye's toolkits, which is based on the motifs identified by Baell et al.[62] The remaining molecules were prepared for docking using DOCK3.7 protocols. Conformational ensembles were capped at 200 conformations per rigid segment and an inter-conformer RMSD diversity threshold of 0.5 Å. Creation of

synthetically accessible libraries was based on commercially available building blocks. Chemical SMARTS patterns were used to retrieve molecules that had suitable functional groups for the intended organic synthesis from Enamine's in-stock building block catalog (https://enamine.net/building-blocks/building-blocks-catalog). Building blocks were coupled in silico to the parent scaffold using reaction SMIRKS patterns[66]. Chemical pattern matching and reagent coupling were performed with in-house scripts based on OpenEye's OEToolkits (version 2020.2). The product molecules were filtered using OpenEye's PAINS-filter, and the remaining compounds were prepared for docking using DOCK3.7 protocols as described above. Chemical pattern matching for the three additional protein targets (SMYD3, NUDT5, and PHIP) was performed with OpenEye's OEToolkits and Enamine REAL space database (version February 2024). Both random molecules and fragment elaborations were prepared for molecular docking using the protocols and parameters described above.

## Construction of enumerated chemical spaces

The UniverseGenerator software for generating constrained chemical spaces was developed using the C++ programming language and libraries from RDKit[63], adhering to the C++17 standard. To generate a chemical space of superstructures of a common subgraph, molecules were constructed by attaching substituent libraries to a scaffold of interest according to the following three-step procedure:

**Step 1: Substituent libraries.** Our open-source implementation of the Generated Database (GDB) algorithm[14] was used to enumerate all chemically stable molecules (up to 11 heavy atoms) that are composed of H, C, N, O, S, and halogen atoms (Supplementary Fig. S4). For a more detailed description of this workflow, we refer to the original publication that describes the GDB algorithm[14]. Briefly, planar graphs with N (1 to 11) nodes with at most four other connections (tetravalent carbon atoms) are first generated using the *nauty* toolkit[67]. The generated graphs are converted into saturated hydrocarbons following atomic valency rules and subsequently embedded into three-dimensional conformers using either *corina* or ChemAxon's *molconvert* toolkits[68]. Hydrocarbons with strained conformations are systematically filtered out by first calculating all volumes defined by the all-carbon tetrahedrons present in the molecule. Molecules that have at least one atomic volume smaller than $0.145\ Å^3$ are discarded. Carbon-carbon bonds in hydrocarbons with relaxed conformations were systematically analyzed for their adjacent hydrogens to determine whether a double or triple bond could be introduced. All possible molecules with unsaturations are then constructed via a combinatorial enumeration that modifies viable bonds. Each carbon atom, together with its bound hydrogen atoms, in (unsaturated) hydrocarbons is analyzed to determine whether it can be mutated into a nitrogen or oxygen atom based on valency rules. Molecules with multiple heteroatoms, are constructed via combinatorial enumeration of viable carbon atoms. Molecules are further decorated by chemically transforming existing functional groups in a combinatorial manner, *e.g.*, transformation of carboxylic acids into sulfonic acids and phenolic hydroxyl groups into halogen atoms. The resulting library of decorated molecules encompasses candidate substituents for subsequent superstructure generation. Each compound in this library was prepared for connection onto a scaffold using activation-tags. Substituents could be attached through five distinct mechanisms: introduction of a single, double, or triple bond substituent, fusion of two ring structures, and the spiro-cyclization of two ring structures (Supplementary Fig. S4). To determine if a site (atom or bond) in the substituent is viable for a particular connection mechanism, the available hydrogens are analyzed. Upon identification of a viable connection site, an activation tag (represented by a specific transition metal corresponding to the type of connection, *e.g.*, lutetium for single bonds, hafnium for double bonds, tantalum for triple bonds) is

attached to the substituent, and the corresponding SMILES string is stored. Each SMILES string is also categorized based on the number of heavy atoms in the substituent and the chemical nature of the connection site, *e.g.*, activated alcohols with five heavy atoms or activated amines with four heavy atoms. This prevents the construction of functional groups that do not abide by the rules of the GDB, *e.g.*, formation of hydrazines, peroxides, and allenes. To accelerate the retrieval of activated substituents, a symmetry analysis of each substituent's molecular graph is carried out prior to activation. Only connection sites that are unique in terms of molecular symmetry are considered for activation, which avoids generation of duplicate activated substituents.

**Step 2: Scaffold activation.** Whereas substituents contain a single activation tag, activated scaffolds can bear multiple activation tags (sites of connection). Analogous to the activation of substituents, all possible connections sites are determined for a given scaffold. These sites are then combinatorially enumerated to find all ways substituents can be attached to the scaffold (Supplementary Fig. S5). The SMILES strings of all activated scaffolds are deduplicated and stored.

**Step 3: Generation of scaffold superstructures.** Each scaffold with K activated connection sites (generated in Step 2) is subjected to the following procedure. A symmetry analysis of the scaffold's molecular graph is carried out to identify connection sites that are identical in terms of molecular symmetry. This step speeds up the algorithm and prevents generation of duplicate superstructures. To construct superstructures of the scaffold that have N additional heavy atoms, an atom-distributing algorithm determines all possible combinations that introduce heavy atoms at K activated connection sites (Supplementary Fig. S6). Each activated connection site must at least receive one new heavy atom because the configuration of zero additional heavy atoms corresponds to a separate activated scaffold configuration where that specific connection site has no activation tag. The sum of newly introduced heavy atoms over the K activated connection sites must be equal to N. To connect new substituents onto a connection site, a SMARTS representation of the connection site is used to retrieve compatible activated substituent libraries (Supplementary Fig. S6). These compatibilities ensure that the formed superstructures comply with GDB rules that also govern the substituent spaces, leading to generation of molecules are deemed relevant and chemically stable.

## Analysis of enumerated chemical spaces

To estimate the sizes of chemical spaces as a function of the number of heavy atoms, logarithmic curves were fitted based on the number of molecules generated by our enumeration algorithm[15]. The curves were fitted using Python's SciPy library.

## Crystallization

Aliquots of purified mOGG1 (22 mg/mL) were pre-incubated individually with the compounds **1, 2, 7, 17** (12.5 mM), **5** (6.25 mM), **8** (1.25 mM), **3** (6.5 mM), or **4** (6.5 mM). All protein samples were crystallized via sitting drop vapor diffusion in various conditions of Morpheus Screen (Molecular Dimensions). This included 0.09 M Halogens, 0.1 M Buffer System 1 pH 6.5, 30.0% v/v GOL_P4K (mOGG1-compound **1**), 0.12 M Monosaccharides, 0.1 M Buffer System 2 pH 7.5, 30.0% v/v GOL_P4K (mOGG1-compound **2**), 0.12 M Alcohols, 0.1 M Buffer System 1 pH 6.5, 30.0% v/v GOL_P4K (mOGG1-compound **5**), 0.09 M Halogens, 0.1 M Buffer System 1 pH 6.5, 30.0% v/v P500MME_P20K (mOGG1-compound **7**), 0.12 M Alcohols, 0.1 M Buffer System 2 pH 7.5, 30.0% v/v EDO_P8K (mOGG1-compound **8**), 0.09 M Halogens, 0.1 M Buffer System 1 pH 6.5, 37.5% v/v MPD_P1K_P3350 (mOGG1-compound **17**), 0.03 M ethylene glycols, 30% v/v GOL_P4K, 0.1 M MOPS/HEPES-Na pH 7.5 (mOGG1-compound **3**), or 0.02 M monosaccharides, 30% v/v GOL_P4K, 0.1 M MES/imidazole pH 6.5 (mOGG1-compound **4**) at 18 °C.

Protein crystals were fished without additional cryoprotectant and flash-frozen in liquid nitrogen.

## Data collection, structure determination, and refinement
X-ray diffraction data was collected at stations I24 (mOGG1-compound **1** and mOGG1-compound **2**), I03 (mOGG1-compound **7** and mOGG1-compound **17**), I04 (mOGG1-compound **3** and mOGG1-compound **4**) of the Diamond Light Source, the BioMAX beamline at MAXIV (mOGG1-compound **8**) and the P13 beamline at PETRA3 (mOGG1-compound **5**). Complete datasets were collected on single crystals at 100 K for each complex. All datasets were processed and scaled with xia2[69], DIALS[70] and Aimless[71] within the CCP4 suite[72]. Molecular replacement was performed in Phaser[73] using the structure of mouse OGG1 (PDB ID: 6G3Y) with all ligands and waters removed, as the search model. Several rounds of manual model building and refinement were performed using COOT[74] and REFMAC5[75] during which waters and ligands were incorporated into the structures.

## OGG1 glycosylase activity assay
The OGG1 Glycosylase Assay was performed in black 384-well plates (OptiPlate-384-F, 6007279 PerkinElmer) using final concentrations of 25 mM Tris-HCl pH 8.0, 15 mM NaCl, 2 mM MgCl$_2$, 0.5 mM DTT, 0.0022% Tween-20, and 1:1000 dilution of dialyzed fish gelatin (Sigma G7765), 41 pM human OGG1 enzyme, 2 nM APE1 and 10 nM 8-oxoA:C substrate in a final volume of 50.5 μL. The 8-oxodA:C substrate was a duplex oligonucleotide where 5′- FAM-TCTG CCA 8CA CTG CGT CGA CCT G-3′ was annealed to a 25% surplus of 5′- CAG GTC GAC GCA GTG CTG GCA GT-Dab-3′ to make sure the annealed DNA is quenched. "8" signifies 8-oxoA and "FAM" and "Dab" signify fluorescein and dabcyl (TriLink Biotech). Briefly, compounds dissolved in DMSO were dispensed using an Echo 550 (Labcyte), followed by transfer of enzyme and substrate solutions manually with a 16-channel pipette or with a Multidrop (Thermo Scientific). The plates were centrifuged, sealed and incubated at room temperature overnight (15 h) and read the following morning in an Hidex Sense plate reader (Hidex Oy) using a 485-nm filter with a bandwidth of 10 nm for excitation and a 535-nm filter with a bandwidth of 20 nm for emission.

## Base excision repair enzymes and NUDIX hydrolase selectivity assays
Inhibition of the following enzymes was assessed in the selectivity assays: apurinic/apyrimidinic endonuclease (APE1), endonuclease 8-like 1 (NEIL1), oxidized purine nucleoside triphosphate hydrolase (MTH1) and single-strand selective monofunctional uracil DNA glycosylase (SMUG1). BER enzymes were assayed with a similar strategy as OGG1, using identical DNA sequences surrounding the substrate lesion and the same reaction buffer containing 2 nM APE1. To assess SMUG1 inhibition, a 375 nM substrate containing uracil opposite guanine and 0.3 U SMUG1 enzyme (M0336 from New England Biolabs) was used. To assess NEIL1 inhibition, a 20 nM substrate containing thymidine glycol opposite adenine and 10 nM NEIL1 was used. To assess APE1 inhibition, an UNG2 substrate was pre-treated with *E. coli* Uracil-DNA glycosylase (New England Biolabs M0280) to generate an AP-site opposite adenine, which was used at 10 nM concentration in the presence of 0.1 nM APE1. All BER inhibition assays were in the linear range and less than 40% of the total substrate had been consumed at readout. The experiments were performed in three independent replicates and the percentage inhibition was calculated relative to the signal of a DMSO-treated control. To assess MTH1 inhibition, MTH1-catalyzed dGTP hydrolysis was coupled to inorganic pyrophosphatase (PPase), thereby releasing inorganic phosphate. The produced free phosphate was then detected using the malachite green assay[76]. These conditions gave robust assays with Z′-factors between 0.5 and 1, and signal to background ratios above 3. Substrate concentration at the $K_m$ value for the respective substrate was chosen if possible. Inhibition using 100 μM compound was tested in reaction buffer (100 mM Tris-Acetate pH 8.0, 10 mM magnesium acetate, 40 mM NaCl, 1 mM DTT, and 0.005% Tween-20). Samples were incubated with shaking at 22 °C for 15 minutes. The reaction was then stopped by the addition of the malachite green reagent[76]. Absorbance was read at 630 nm using a Hidex Sense plate reader after a period of incubation with the methylene green reagent of 15 minutes. The experiment was performed in three independent replicates and percentage inhibition was calculated relative to the signal of a DMSO-treated control.

## Differential scanning fluorimetry
The differential scanning fluorimetry (DSF) assay was performed in white 384-well plates (04729749001, LightCycler 480 Multiwell Plate, Roche Diagnostics) using final concentrations of 25 mM Tris-Acetate pH 7.5, 50 mM CaCl$_2$, 10% glycerol, 1 mM DTT, 4 μM human OGG1 protein, 5× SYPRO Orange dye (S6651, Invitrogen, Ex492/Em610) in a final volume of 10.1 μL/well. Briefly, compounds dissolved in DMSO were dispensed using an Echo 550 (Labcyte), followed by transfer of enzyme mix manually with a 16-channel pipette. The plates were sealed (04729757001 LightCycler480 Sealing Foil, Roche Diagnostics), centrifuged and read in a LightCycler® 480 (Roche Diagnostics). The temperature was increased from 20 °C to 85 °C with 10 readings/degree. $T_m$ values were calculated using an excel template and fitting to the Boltzmann equation in Prism v.6.07 (GraphPad Software).

## Cellular thermal shift assay
CETSA experiments were carried out with intact HL60 cells treated in culture. Cells were seeded at a density of 350 000 cells/mL and treated with DMSO (0.01%, v/v) or 20 μM test compound for 2 h at 37 °C and 5% CO$_2$ in a humidified incubator. Cells were washed once in 1x TBS (50 mM Tris-HCl, pH 7.6, 150 mM NaCl), resuspended in 1x TBS supplemented with protease inhibitors (Roche), and then divided into 30-μL aliquots corresponding to approximately 600,000 cells per sample in PCR strip tubes. Cells were heated in a Veriti Thermal Cycler (Applied Biosystems) for 3 min at indicated temperatures (temperature interval consisting of 2 °C increases, starting from 40 °C up to 62 °C) and cooled for 5 min at room temperature to allow precipitation of denatured proteins. 5x NP-40 buffer was added to the samples and cells were lysed by performing three freeze–thaw cycles at −80 °C for 3 min and at 37 °C for 3 min with gentle vortexing between the cycles. Cell lysates were then centrifuged at 17,000 $g$ and 4 °C for 30 min to remove cellular debris and insoluble proteins. Supernatant was prepared for western blot analysis according to standard procedures. Briefly, 14 μL of the total protein samples were loaded on RunBlue 4-12% Bis-Tris gels (Westburg) using 4x NuPage LDS Sample buffer (Thermo Fisher Scientific) supplemented with 10 mM DTT (Sigma Aldrich) and blotted onto nitrocellulose membrane according to standard protocols. Membranes were blocked with 5% milk-powder in TBST and incubated with the indicated primary antibodies (α-OGG1 (Abcam, Cat. #ab124741, 1:1000) and anti-mouse SOD1 (G-11) (Santa Cruz Biotechnology, Cat. #sc-17767, 1:1000) overnight at 4 °C. Incubation with secondary antibody was performed in light-protected conditions for 2 h at room temperature. Detection was performed with the Odyssey CLx Infrared Imaging System (LI-COR). ImageJ was used for image analysis.

## Laser microirradiation assay
Assays measuring recruitment of OGG1-GFP to laser-induced DNA damage sites in U2OS cells treated with **TH5487** and compound **23** were carried out as described previously[77]. In brief, U2OS cells stably expressing GFP-tagged wild-type OGG1 were seeded on Ibidi μ-dish (Ibidi #81166) 24 h prior to the indicated treatment. For laser microirradiation, cells were pre-sensitized with 10 μg/mL Hoechst 33342

 

(Thermo Fisher Scientific, Catalog No. 62249) for 10 min at 37 °C. To avoid background fluorescence from phenol red present in the DMEM culture medium, we exchanged the medium to live cell imaging medium (Thermo Fisher Scientific, Catalog No. 31053028) supplemented with penicillin−streptomycin antibiotics, 10% FBS and 25 mM HEPES containing either 0.1% DMSO or 10 μM **TH5487** or compound **23** for 1 h. Cells were then transferred to a 37 °C pre-heated environmental chamber attached to a Zeiss LSM 780 confocal microscope equipped with a UV-transmitting Plan-Apochromat 40x/1.30 Oil DIC M27 objective. To induce DNA damage, a nuclear spot (dimensions: 10 x 10 pixels) was selected using the circular region tool of the ZEN software (ZEN, Zeiss, Germany) and irradiated using a 405 nm diode laser set to 100% (spot irradiation, 1 iteration, zoom 5, and pixel dwell time of 12.61 μs). For quantitative evaluation of the recruitment kinetics, the fluorescence intensity at the irradiated spot was corrected for background and for total nuclear loss of fluorescence over the time course and normalized to the pre-irradiation value.

### NF-κB assay

The inhibition of TNF-α induced NF-κB activation was measured in HEK293T cells transfected with the pGreenfire NF-κB plasmid. This is a plasmid that has NF-κB response elements (TRE) coupled to GFP and luciferase expression. When the NF-κB is activated by TNF-α, it binds to the TRE and induces GFP expression, and the fluorescence can be measured in a plate reader. The cells were authenticated with STR profiling, frozen down and these stocks were used in experiments. The HEK293T pGF NF-κB cells were kept in DMEM with 5% FBS and 1% P/S but, when seeded for the assay the medium was changed to DMEM FluoroBrite with 1% dialyzed FBS. All cell culture reagents were purchased from Thermo Fisher Scientific. Compounds were nano-dispensed (Echo acoustic dispenser) in 96-well, black/clear, assay plates (Corning #3904), duplicate plates, in 8-points dose-response curves, 1:2 dilutions, with DMSO compensation (0.3% DMSO). Cells were seeded at 30000 cells/well in medium containing 10 ng/mL of TNF-α (Sigma-Aldrich), except negative control wells which only contained cells in medium. Cells were cultured for 24 hours in a humidified incubator at 37 °C with 5% CO$_2$ after which the GFP fluorescence was measured in a Sense plate reader (Hidex) with ex485/em535. The RFU data from the duplicate plates were combined to calculate % inhibition of NF-κB activation and IC$_{50}$-values were calculated using XLfit (IDBS). In parallel with the NF-κB assay, a CellTiterGlo (Promega) cell viability assay was run to exclude compounds affecting cell viability. Compounds were nano-dispensed (Echo acoustic dispenser) in assay plates in 11-points dose-response curves, 1:3 dilutions, without DMSO compensation, with duplicates on the same plate. Cells were seeded at 7500 cells/well in 25 μL medium (DMEM FluoroBrite with 1% dialyzed FBS) containing 10 ng/mL TNF-α and cultured for 24 hours, with column 24 containing only culture medium. After equilibration to room temperature, the CellTiterGlo reagents were mixed and 25 μL/well were added to the cell plate. After 2 min mixing of the plate on an orbital shaker, the plate was incubated for 10 min at room temperature followed by luminescence reading in Sense. The cell viability was calculated compared to untreated control wells and the EC$_{50}$ values were determined in XLfit (IDBS). Three independent experiments were carried out and for each experiment, five individual cells were recorded.

### Cancer cell viability assay

Three different cancer cell lines A2780 (ovarian cancer, ECACC 93112519, female), A549 (lung cancer, ATCC 60150896, male) and HCT116 (colon cancer, ATCC CCL-247, male), and one immortalized fibroblast cell line (BJhTERT) as control, were seeded in black/clear 384-well plates (Corning #3764), 50 μL/well, containing 11 concentrations, nano-dispensed, dose-response curves of the test compounds. A2780, HCT116, and HEK293T cell lines were authenticated with STR

profiling, frozen down, and these stocks were used in experiments. The A2780 cells were cultured in RPMI 1640, the A549 and BJhTERT cells in DMEM, and the HCT116 cells in McCoys 5 A medium. All media were supplemented with 10% FBS and 1% P/S. All cell culture media, additives, and reagents were purchased from Thermo Fisher Scientific. The cell plates were incubated 72 hours in a humidified incubator at 37 °C with 5% CO$_2$, after which resazurin (Sigma-Aldrich), 10 μL/well, was added to a final concentration of 10 μg/mL. The plates were incubated for 6 hours with resazurin followed by fluorescence readings in a Sense plate reader (Hidex), resorufin protocol (ex 544 nm/em 595 nm). The cell viability was calculated compared to untreated control wells and the EC$_{50}$ values were determined in XLfit (IDBS).

### Metabolic stability in the presence of human liver microsomes

Human liver microsomes were purchased from BioIVT. Metabolic stability was determined in 0.5 mg/mL human liver microsomes at a compound concentration of 1 μM in 100 mM KPO$_4$ buffer pH 7.4 in a total incubation volume of 500 μL. The reaction was initiated by the addition of 1 mM NADPH. At various incubation times, i.e., at 0, 5, 10, 20, 40, and 60 min, a sample was withdrawn from the incubation and the reaction was terminated by the addition of cold acetonitrile with warfarin as an internal standard. The amount of parent compound remaining was analyzed by liquid chromatography coupled to triple quadrupole mass spectrometry (LC-MS/MS).

### Plasma protein binding and plasma stability in human plasma

Pooled human plasma was provided by Uppsala Academic Hospital and was collected from two donors (non-smoking) (citric acid). In brief, 0.2 mL of the plasma (50% plasma, 50% isotonic buffer) test solution (typically 10 μM final compound concentration) was transferred to the membrane tube in the RED insert (ThermoFisher Scientific). 0.35 ml isotonic phosphate buffer pH 7.4 was added to the other side of the membrane. The 96-well base plate was then sealed with an adhesive plastic film (Scotch Pad) to prevent evaporation. The sample was incubated with rapid rotation (>>900 rpm) on a Kisker rotational incubator at 37 °C for 4 h to achieve equilibrium. Prior to LC-MS/MS analysis the plasma and buffer samples were treated with the addition of Methanol (1:3) containing Warfarin as an internal standard to precipitate proteins. The standard curve was created using the plasma standard. The plate was then sealed, and centrifuged and the supernatant was analyzed by liquid chromatography coupled to triple quadrupole mass spectrometry (LC-MS/MS).

### Caco-2 cell permeability assay

Caco-2 cell monolayers (passage 94-105) were grown on permeable filter support and used for transport study on day 21 after seeding. Prior to the experiment a drug solution of 10 μM was prepared and warmed to 37 °C. The Caco-2 filters were washed with pre-warmed HBSS prior to the experiment, and thereafter the experiment was started by applying the donor solution on the apical or basolateral side. The transport experiments were carried out at pH 7.4 in both the apical and basolateral chamber. The experiments were performed at 37 °C and with a stirring rate of 500 rpm. The receiver compartment was sampled at 15, 30, and 60 min, and at 60 min also a final sample from the donor chamber was taken to calculate the mass balance of the compound. The samples (100 μL) were transferred to a 96-well plate containing 100 μL methanol and Warfarin as IS and was sealed until analyzed by liquid chromatography coupled to triple quadrupole mass spectrometry (LC-MS/MS).

### Liquid chromatography coupled to triple quadrupole mass spectrometry (LC-MS/MS)

The test compounds were optimized on a Waters Acquity UPLC XEVO TQ-S microsystem (Waters Corp.) operating in multiple reaction monitoring (MRM) mode with positive or negative electrospray

ionization. Compounds were optimized by using the QuanOptimize software (Waters Corp.). The following MS conditions were used:

| Transition m/z | Dwell time (s) | Cone voltage | Collision energy |
|---|---|---|---|
| 324.4 > 128.04 | 0.028 | 10 | 60 |
| 324.4 > 171.03 | 0.028 | 10 | 50 |

For chromatographic separation, a $C_{18}$ BEH 1.7 μm column was used with a general gradient of 5% to 1000% of mobile phase B over a total running time of 2 min. Mobile phase A consisted of 0.1% formic acid in purified water, and mobile phase B of 0.1% formic acid in 100% acetonitrile. The flow rate was set to 0.5 mL/min and 5 μL of the sample was injected.

### Reporting summary

Further information on research design is available in the Nature Portfolio Reporting Summary linked to this article.

### Data availability

The ZINC15 library is available at https://zinc15.docking.org. The PDB entry for the mOGG1 crystal structure used for molecular docking calculations is 6G3Y. All compounds tested are listed in in the Supplementary Information and Source Data File. Chemical identities, purities (LC/MS), yields and spectroscopic analysis ($^1$H,$^{13}$C NMR) for active compounds are provided in Supplementary Information. The crystallographic data generated in this study have been deposited in the PDB database under accession codes 7QEL, 7ZG3, 8CEX, 8CEY, 7Z5R, 7ZC7, 7Z3Y, and 7Z5B. Source data are provided with this paper.

### Code availability

The UniverseGenerator source code is freely available and can be found on the following GitHub repository (https://github.com/carlssonlab/UniverseGenerator). The original code has deposited on Zenodo (https://zenodo.org/records/14460126). Other scripts to process molecular docking results can be found on the following GitHub repository (https://github.com/carlssonlab/frag2lead).

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

## Acknowledgements

A.L. was supported by a postdoctoral scholarship of the Knut and Alice Wallenberg Foundation (KAW2022.0347). J.C. received funding from the European Research Council (ERC) under the European Union's Horizon 2020 research and innovation programme (grant agreement: 715052), the Swedish Cancer Society (22 2473 Pj), the Swedish Strategic Research Program eSSENCE, and the Swedish Research Council (2021-03464). This work was supported by grants from the Swedish Research

Council (2015-00162) and the Swedish Cancer Society (24 3694 Pj) to T.H. This work was supported by grants from the Swedish Research Council (2022-03681) and the Swedish Cancer Society (20 1287 PjF) to P.S. The computations were enabled by resources provided by the National Academic Infrastructure for Supercomputing in Sweden (NAISS), partially funded by the Swedish Research Council through grant agreement no. 2022-06725, and the Berzelius resource provided by the Knut and Alice Wallenberg Foundation. A.L. and J.C. thank OpenEye Scientific Software for the use of OEToolkits at no cost. E.S., J.D. and S.K. thank the beamline scientists at Diamond Light Source (UK), MAXIV (Sweden) and PETRA3 (Germany) for their support with X-ray diffraction data collection. The authors thank the Uppsala University Drug Optimization and Pharmaceutical Profiling Platform (UDOPP) for the determination of in vitro pharmacokinetic properties. This project was supported by the Compound Center at the Chemical Biology Consortium Sweden (CBCS). CBCS is funded by the Swedish Research Council (2021-00179) and SciLifeLab. This project has received funding from the Innovative Medicines Initiative 2 Joint Undertaking (JU) under grant agreement No 875510 (M.M., E.H., and E.W.). The JU receives support from the European Union's Horizon 2020 research and innovation programme and EFPIA and Ontario Institute for Cancer Research, Royal Institution for the Advancement of Learning McGill University, Kungliga Tekniska Högskolan, Diamond Light Source Limited. This article reflects the views of the authors and the JU is not liable for any use that may be made of the information contained herein.

## Author contributions

A.L. and J.C. designed the study. A.L. performed the molecular docking calculations, wrote code for generating and analyzing chemical spaces, and designed compounds under the supervision of J.C. A.L. and F.B. performed substructure and similarity calculations. D.D.V. performed compound synthesis under the supervision of J.K. E.R.S, J.D., S.K, and G.M. determined crystal structures of inhibitors under the supervision of P.S. Biological assays were performed under the supervision of T.H. E.W. performed and analysed biochemical experiments. I.A. performed and analysed cellular assays. L.M and M.L performed and analysed target engagement assays under the supervision of O.M. O.W., M.S., E.H., M.M, C.K, and U.W.B. contributed to coordination of the project, acquisition of reference compounds for the assays, data analysis, and compound synthesis. A.V.T., D.S.R., and Y.S.M. provided the Enamine REAL space and analytical data for the synthesized compounds. A.L. and J.C. wrote the manuscript with contributions from the other authors.

## Funding

## Competing interests

J.C. is a founder of DareMe Drug Discovery Consulting. O.W. and T.H. are listed as inventors on U.S. patent no. WO2019166639 A1, covering a different class of OGG1 inhibitors than those described in this work. The patent is fully owned by a non-profit public foundation, the Helleday Foundation, and T.H. is a member of the foundation board. U.W.B., C.K. and M.S. are employees of Oxcia, a company developing OGG1 inhibitors towards the clinic. T.H. is a member of the board of Directors of Oxcia. D.S.R. and Y.S.M. are employed with Enamine Ltd. Y.S.M. serves as a scientific advisor to Chemspace LLC. The remaining authors declare no competing interests.
