## [Transparent Peer Review file · Nature Communications]

Virtual Fragment Screening for DNA Repair Inhibitors in Vast Chemical Space

Corresponding Author: Professor Jens Carlsson

Version 0:

Reviewer comments:

Reviewer #1

(Remarks to the Author)

The manuscript entitled; "Virtual Fragment Screening for DNA Repair Inhibitors in Vast Chemical Space" used a computational docking approach to evaluate extremely large chemical libraries with OGG1 as the target. Overall, the study is interesting and well described from an established group. From what is described the study is scientifically accurate, with a few minor comments listed below. The problem I have with the study is the impact to the field. The authors didn't identify a better compound for OGG1 than TH5675 and limited the testing of their approach to OGG1. I recognize the importance of having a controlled system. However, this results in the major impact of the study being the narrative that the authors screened a vast number of compounds and chemical space. While this is true, it isn't evident how this helped create a better compound or how this could be used for a novel target. While it is very likely this would work for a novel target, the authors don't show this. While not required for publication in other journals, I do feel this is necessary for Nature Communications. Finally the ability for this to be employed by other groups is very unlikely and this reduces impact. The authors often go from a very large number of compounds to a smaller more manageable group through their judgment and guidance of the TH5675 compound/binding site. They clearly have the expertise to do this, but providing more details on how they did this would help make the approach more applicable to others and drive impact to the field.

1. The authors state, "A set of 29 compounds was selected for experimental evaluation from the 500 top-ranking clusters based on visual inspection of their complementarity to the active site." The authors need to add more description on how they went from 500 to 29 so future researchers could implement this approach.
2. How did the authors identify the "500 top ranking clusters"?
3. The xray table has several issues that make it difficult to evaluate. It appears 8CEY is reported slightly differently in regards to the way the units are presented (e.g. Rmerge, resolution). The table should have the same way of reporting the different structures. What are the outliers in 7Z5R the authors are referring to?
4. Why did the authors select the 99 and 495uM concentrations for the DSF experiments? In the DSF experiments did any of the other 65 compounds induce a stabilization other than 1 and 2? Why did the authors only focus on these two.
5. The figure panels nomenclature is distracting in that the order from a-f is not always in one direction. Compare figure 3 vs 4.

(Remarks on code availability)

Reviewer #2

(Remarks to the Author)

In this manuscript, Lutten and colleagues report new inhibitors of the OGG1 DNA-repair protein, a prominent and not yet thoroughly exploited oncotarget. Starting from a docking-based ultrahigh-throughput virtual screening (uHTVS) campaign of 14M fragments (vs. a control campaign of 235M lead-like compounds), they first identify fragment hits, then characterize their binding modes via X-ray crystallography, then use the resulting structural information in further iterations of uHTVS-based hit-to-lead development, finally resulting in a submicromolar OGG1 inhibitor, demonstrated to be effective in cellulo. Besides the already important discovery of this compound, the paper is a valuable contribution to the growing toolbox of uHTVS

methods, a definitive trend of the past few years of computer-aided drug discovery. Specifically, the authors use the structural information gathered during the first fragment screening to narrow down the commercially available chemical space and/or reagent-based, synthesizable chemical space during the subsequent iterations, based on the scaffold of the original fragment hit, as well as its orientation within the binding site (and its effect on possible growing vectors). Therefore, despite the lead-like size range, the follow-up chemical space reaches the range which is feasible for docking in full, and no sampling is required (where normally a minuscule portion of the chemical space is docked). The authors publish the code for generating the relevant chemical spaces via github (UniverseGenerator), which can be subsequently used for the uHTVS calculations. I am convinced that this is a valuable methodology that is suitable for publication in Nat Comm, as to my knowledge this is the first time that a uHTVS workflow (as well as the generated chemical space) is tailored specifically to fragment-to-lead elaboration.

Nonetheless, the authors should improve the discussion on the added value of their methodology with respect to recent workflows that apply the concept of chemical space generation via the combination of commercially available reagents, such as V-SYNTHES (<https://www.nature.com/articles/s41586-021-04220-9>) or SASS (<https://doi.org/10.1021/acs.jcim.3c01865>).

The code should be improved as well, in terms of the scripts' access to the RDKit library, see below in the dedicated section of remarks.

There are a few more minor points to address as well:

- Fig 1c: it is confusing that the authors highlight the 11B commercial space when it was not yet applied/considered at this step (initial fragment screen).
- Fig 2d: how many compounds are there in the tailored chem space? What is the number of reagents?
- Fig 4f: the caption erroneously specifies compound 13 instead of compound 17.

(Remarks on code availability)

The code is a collection of logically structured scripts, supplemented with a demo dataset and a thoroughly written README file that guides through the installation of the package, as well as its dependencies.

During installation though, I had to adjust parts of the code to get the desired result:

- it is correctly "conda create -n universe", instead of "conda env create -n universe"
- I failed to install RDKit from source ("make" step, error during the installation of the dependency "clang"), so I installed it via conda. However, due to different cross-dependencies, I had to try different versions of python in my environment until I arrived at 3.10, where both boost and rdkit could be installed. The authors should specify python 3.10 as a working version for this route.
- In the installation route above, "export RDBASE=\${UniverseGenerator}/rdkit" will not necessarily work, e.g. for me, the base directory was "/opt/anaconda3/envs/universe/include/rdkit"
- The script "build-instructions" is erroneously referred as "generate-instructions" in the README

Further on, I ran into issues regarding the scripts' access to the RDKit library:

- the script "adjmat2smiles" will look at a fixed path to check for RDKit libraries, e.g. "/usr/local/lib/libRDKitGraphMol.1.dylib", "/usr/local/lib/libRDKitOptimizer.1.dylib", etc. and result in an error if RDKit is installed elsewhere (e.g. via conda). One can try to create symbolic links to rectify this but it's tedious for all dylibs and will not necessarily solve the issue. Same thing happens with "eliminate-strained", "unsaturate", "functionalize", "decorate", "activate-substituents", "activate-scaffold", "build-instructions" and others.
- Note that this is unaffected by the RDBASE variable set above.

Overall, the steps that I could reproduce have worked properly and I'm willing to accept that the scripts are working if the environment/dependencies are set up exactly in the way the authors have on their machines. However, the authors should make the code a bit more robust in terms of the scripts' access to the RDKit library, which seems to be heaviest dependency of the package.

Reviewer #3

(Remarks to the Author)

Lutens and co-workers have performed a large fragment-based drug discovery (FBDD) study where they have first identified fragments that bind to a promising oncology/inflammation target called OGG1 using structure-based virtual screening. They verified these experimentally using X-ray crystallography. Then, the weak fragments were optimized into sub-micromolar inhibitors suitable for further development by docking an enumerated virtual library to the new crystal structures. In addition to this, the authors studied on how to do the virtual screening using virtual on-demand libraries most efficiently in the context of FBDD.

This study is highly significant for several reasons: firstly, there are only few OGG inhibitors reported in the literature and this study contributes new chemical structures together with crystal structure that can be used by others as a starting point to design new inhibitors for this interesting target. Secondly: quoting the authors "In contrast to common belief, our results support the use of molecular docking as a tool to identify fragment ligands.". Indeed, this study is an impressive success-story of using molecular docking in FBDD. Finally, they provide important insights on the fragments and virtual chemical spaces plus a concrete tool UniverseGenerator to investigate large chemical spaces.

The computationally predicted docked poses for novel compounds were experimentally confirmed using X-ray crystallography with sufficient resolution. The selectivity and cell activity of compounds were investigated to the level that the

claim ending the manuscript is warranted (“attractive leads for further development”). The authors provide the code for UniverseGenerator which I was able to compile successfully, but I was unable to test it properly as couple of input files were missing from the package.

The manuscript is well-written and cites the relevant literature. The experimental procedures performed in this study (crystallization, compound synthesis, assays) are beyond my expertise and I am unable to comment on those in detail. I have only the following minor suggestions regarding the computational part that might improve the manuscript.

1) “The obtained control sets were used to evaluate the performance of the docking grids by means of ligands-over-decoys enrichments. Enrichment values and predicted poses of ligands were used to select the final grid parameters.” (Page 24). The retrospective virtual screening performance of the final grid selected for prospective screening should be described in more detail (for example ROC curves could be added to Supporting Information, see <https://www.ncbi.nlm.nih.gov/pmc/articles/PMC4631717/>).

2) “Chemical pattern matching and reagent coupling were performed with in-house scripts based on OpenEye’s OEToolkits (version 2020.2).” (Page 26)
The virtual reactions and their SMIRKS patterns should be included in the Supporting Information.

3) Table 1, Crystal structure for THR5487: PDB 6RLW is human OGG1 and not mouse OGG1 like the other crystal structures in this work. Authors should highlight this in the Table 1. It should be noted in the text that the difference between the two proteins is very small as can be seen in Reference 23 (Figure 3A and 3B) so that reader doesn’t have to dig this small piece of information from an another study.

4) “Crystallographic waters and other solvent molecules were removed from the structure, with the exception of the binding site water molecule with residue number 504.” (Page 23)
The reason for keeping this one particular water is unclear to the reader (short additional explanation would be beneficial).

5) “Top-scoring molecules were filtered using a PAINS-filter to reduce the risk of encountering false positives” (Page 24)
“Matching molecules were filtered by a PAINS-filter” (Page 25)
“The product molecules were filtered using a PAINS-filter,” (Page 26)
There are quite a few implementations of the PAINS-filter floating around. The authors should specify the specific version that they used (for example, RDKit version like described here https://projects.volkamerlab.org/teachopencadd/talktorials/T003_compound_unwanted_substructures.html).

6) “The diversity among the top-ranked compounds was increased by clustering the best scoring molecules using ECFP4-based fingerprints and a Tanimoto coefficient of 0.5.” (Page 24)
The toolkit (RDKit?) used to do these tasks should be explicitly mentioned here.

7) “Creation of synthetically accessible libraries was based on commercially available building blocks.” (Page 25)
It is not clear where the commercially available building blocks were extracted here (from ZINC or just from Enamine?).

8) Regarding references, I happened to note that there is a typo in reference #11 (the year should be 2016).

(Remarks on code availability)

The code is not yet publicly in github, so I used the version that was included with the manuscript.

I compiled UniverseGenerator on Ubuntu 22.04 LTS/x86_64 successfully. There was one small bit missing from in the instructions which is described here (<https://greglandrum.github.io/rdkit-blog/posts/2023-03-17-setting-up-a-cxx-dev-env2.html>): user needs to define LD_LIBRARY_PATH by “export LD_LIBRARY_PATH=\$RDBASE/lib:\$CONDA_PREFIX/lib” before running the script “prepare_universe.sh”. This should be added to instructions as otherwise linking will fail.

It was not possible to run the examples in UniverseGenerator as the files “filter_file.tsv” and “smirks_library.tsv” were missing. These would need to be added and these should be the ones that were used to generate data discussed in Pages 16-18.

When trying to run the examples with UniverseGenerator, I also noticed that following things should be fixed in README.md:

Replace:

```
# symbolic link for next step  
ln -s 1.generate/output.embed.mol2 2.strain/input.eliminate.mol2
```

With:

```
# symbolic link for next step  
ln -s ${UniverseGenerator}/demo/1.generate/output.embed.mol2 ${UniverseGenerator}/demo/2.strain/input.eliminate.mol2
```

Replace (note the typo in the 3.unsature):

symbolic link for next step
ln -s 2.strain/output.eliminate.smi 3.unsaturation/input.unsaturation.smi

With:
symbolic link for next step
ln -s \${UniverseGenerator}/demo/2.strain/output.eliminate.smi \${UniverseGenerator}/demo/3.unsaturation/input.unsaturation.smi

Replace:
symbolic link for next step
ln -s 3.unsaturation/output.unsaturation.smi 4.functionalize/input.functionalize.smi

With:
ln -s \${UniverseGenerator}/demo/3.unsaturation/output.unsaturation.smi
\${UniverseGenerator}/demo/4.functionalize/input.functionalize.smi
Replace:
symbolic link for next step
ln -s 4.functionalize/output.functionalize.smi 5.decorate/input.decorate.smi

With:
symbolic link for next step
ln -s \${UniverseGenerator}/demo/4.functionalize/output.functionalize.smi
\${UniverseGenerator}/demo/5.decorate/input.decorate.smi

Reviewer #4

(Remarks to the Author)

Review of Luttens et al Nature Communications

The manuscript by Luttens et al reports a reduction-to-practice strategy for the identification of potent inhibitors of the DNA glycosylase OGG1 through virtual fragment screening algorithms. The study is well-justified since OGG1 is an outstanding target for the suppression of NF κ B-mediated inflammation and cancer therapeutics. Prior investigations on the development of OGG1 inhibitors has identified a promising chemotype, with a biologically potent lead, TH5487 that has an IC₅₀ of 340 nM for the inhibition of the OGG1 glycosylase activity and has shown promising results in multiple biological end-point assays. To improve on these inhibitors, the current study utilized virtual screening and selected syntheses to test predicted efficacy for potential leads. However, enthusiasm for these investigations were greatly tempered by the findings that at the end of the experimental pipeline, the best, newly described molecules had IC₅₀s less potent (600 nM) than the previously described inhibitor. These findings may reflect challenges created by starting with crystallographic data using TH5675 complexed with OGG1. The potential upside of the proposed strategies could be in its general application to the improvement of inhibitors of other enzymes for which crystal coordinates of lead molecules have been resolved at high resolution.

Since the biological studies involving OGG1 inhibitors have almost exclusively focused on TH5487, it is not clear why TH5675 is used as the gold-standard from which to simulate fragment screening. It would seem much more germane to use structural information concerning this complex than that currently chosen.

Given that the starting structure for the docking screens was that of 6G3Y, chain A, and the authors acknowledge that analyses of OGG1 structures reveal that "the active site is highly polar and adopts different shapes in the complexes with the inhibitor and DNA", in the Results section, the authors should provide greater justification for the use of this as their starting structure. Again, it may suggest that the structure of the biologically-relevant inhibitor would be a far-preferable starting structure.

Comparisons of the IC₅₀ data with the Δ T_m values, does not reveal any predictive trends; could the authors speculate on the seemingly lack of correlation of those values? It would seem reasonable that protein stabilization and catalytic inhibition would have a highly significant correlation.

Comparison of data presented in Tables 1 and 2, it is unclear why the authors report IC₅₀ values in Table 1 and pIC₅₀ values in Table 2. The text associated with Table 2 presents the IC₅₀, not the pIC₅₀ determinations. Unless there is a compelling reason to change to pIC₅₀, it would be better to consistently report the IC₅₀ data.

In the text, the authors reference fold improvements over that of their reference compd 1 - given that the IC₅₀ of compd 1 was reported as >99 μ M, how can fold improvements be calculated given that no definitive IC₅₀ was determined? It would seem much more reasonable to assess the success of the screenings based on the previously reported IC₅₀ for TH5487 being 340 nM, with a change in T_m = 4.3.

(Remarks on code availability)

Version 1:

Reviewer comments:

Reviewer #1

(Remarks to the Author)

The authors have addressed all my concerns and the manuscript is suitable for publication.

(Remarks on code availability)

Reviewer #2

(Remarks to the Author)

The authors have revised the manuscript with a thorough consideration of my, and as much as I could judge, the other reviewers' comments and have adequately reflected to the questions/suggestions. Notably, the UniverseGenerator code was significantly improved and is much easier to apply in its current revision. In my opinion, the paper can be published in its present form.

(Remarks on code availability)

The UniverseGenerator package was significantly improved in its current revision and is much easier to use now.

Reviewer #3

(Remarks to the Author)

The authors have responded to all my requests and therefore the manuscript is acceptable for publication in my opinion.

(Remarks on code availability)

I was able to compile the code from reviewer ZIP-file and run the demo example, but noticed couple of typos/bugs which are good to fix before the source code is released to the public:

- Code didn't compile directly in Linux due to case sensitivity in (code/8.Superstructure should be code/8.superstructure). This is not visible in MacOSX as there the filesystem doesn't care about this.

- Typos in README instructions (incorrect filenames):

1)

```
In -s ${demo}/5.decorate/output.decorate.smi ${demo}/6.activate/input.substituents.smi
```

should be

```
In -s ${demo}/5.decorate/output.decorate.smi ${demo}/6.substituents/input.substituents.smi
```

2)

```
${UniverseGenerator}/bin/activate-scaffold -i ${demo}/7.scaffold/input.scaffold.smi -o ${demo}/7.scaffold/output.scaffolds.smi
```

should be

```
${UniverseGenerator}/bin/activate-scaffold -i ${demo}/7.scaffold/scaffold.smi -o ${demo}/7.scaffold/output.scaffolds.smi
```

3)

```
${UniverseGenerator}/bin/build-instructions -i ${demo}/8.superstructure/input.scaffolds.smi -o  
${demo}/8.superstructure/superstructure-instructions.csv -n 15
```

should be

```
${UniverseGenerator}/bin/build-instructions -i ${demo}/8.superstructure/reference.input.scaffolds.smi -o  
${demo}/8.superstructure/superstructure-instructions.csv -n 15
```

4)

```
${UniverseGenerator}/bin/build-superstructures -i ${demo}/8.superstructure/superstructure-instructions.csv -o  
${demo}/8.superstructure/output.superstructures.smi -f ${UniverseGenerator}/auxiliaries/superstructure_file.tsv
```

should be

```
${UniverseGenerator}/bin/build-superstructures -i ${demo}/8.superstructure/superstructure-instructions.csv -o  
${demo}/8.superstructure/output.superstructures.smi -f ${UniverseGenerator}/auxiliaries/superstructure_filter.tsv
```

Reviewer #4

(Remarks to the Author)

The revised submission of Luttens et al is very significantly improved relative to the original submission and has

satisfactorily addressed both my major and minor concerns. Rather than simply argue their case, the revision has appropriately expanded the beneficial (improved) properties of the compounds identified through their fragment-based screening approach. The explanation of the use of the co-crystal structure using the TH5675 was informative and in fact highlights the amount of time and effort that was required to bring this strategy to this point.

(Remarks on code availability)

NA to my research designs.

UPPSALA
UNIVERSITET

SciLifeLab

October 14th, 2024

We want to thank the referees for taking the time to review our manuscript ("*Virtual Fragment Screening for DNA Repair Inhibitors in Vast Chemical Space*", Manuscript: NCOMMS-24-23933) and the many constructive suggestions. We have carefully considered all comments and performed additional experiments and calculations, resulting in a substantially stronger manuscript. As requested by the reviewers, we have included a comparison of our virtual screening approach and compounds to the campaign resulting in the discovery of reference inhibitor TH5487. The new results clearly demonstrate that our technique is more efficient than traditional screening and led to the discovery of inhibitors with improved properties. To further increase the impact of our work, we also demonstrate that the approach can be applied to other drug targets. General guidelines for application of fragment-based virtual screening are provided and we enable other researchers to use our tools.

Please find below point-by-point responses to the referees with their comments in **Blue**. Relevant changes to the manuscript have been marked in **yellow** to facilitate the review process.

Reviewer #1

The first reviewer wrote that "*the study is interesting and well described*". There were two major suggestions on how to enhance the quality of the manuscript: **(1)** Show advantages of our virtual screening approach and compounds compared to previously identified inhibitor TH5487; and **(2)** Provide more information regarding how our virtual screening approach can be applied to other targets. We have addressed these important points as well as all the minor comments from the reviewer:

Major comments:

1. Reviewer: "*The authors didn't identify a better compound for OGG1 than TH5675 and limited the testing of their approach to OGG1. I recognize the importance of having a controlled system. However, this results in the major impact of the study being the narrative that the authors screened a vast number of compounds and chemical space. While this is true, it isn't evident how this helped create a better compound or how this could be used for a novel target. While it is very likely this would work for a novel target, the authors don't show this.*"

We agree with the reviewer that it is important to present advantages of our compounds compared to previously identified inhibitors and that the same technique can be applied to other drug targets. In the revised manuscript, we have addressed these questions by comparing to the discovery of OGG1 inhibitor TH5487 and by performing extensive molecular docking calculations for three other drug targets.

• **Comparisons to previous screens and inhibitors of OGG1.** To assess the efficiency of our structure-based fragment screening technique, we compared our results to the discovery of inhibitor TH5487 by Visnes *et al.* (ref. 23), which was based on traditional high-throughput screening and in-house medicinal chemistry efforts. Notably, our fragment docking resulted in >1000-fold higher screening hit rate and identified four diverse starting points for optimization compared to a single scaffold from the high-throughput screen. Moreover, our study involved experimental testing of >150-fold fewer compounds: >18,100 compounds were evaluated in the HTS campaign, and we tested only 107 compounds to discover our most potent inhibitors. This comparison demonstrates the efficiency of our approach and has been added in the discussion of the revised manuscript (pages 24-25).

Another advantage of a fragment-based approach compared to traditional screening is that the resulting lead compounds often have higher quality. In the field of drug discovery, it is widely recognized that the quality of a compound should not solely be judged based on potency. In the hit-to-lead optimization phase, achieving good physicochemical and pharmacokinetic properties is equally important (see ref. 38). As our most potent compounds (**17** and **23**) and TH5487 have comparable activity (TH5487 has a 2-fold better IC₅₀ value in enzyme assay - our inhibitors have a 2-fold better stabilization than TH5487 in the DSF assay), we evaluated physicochemical (lipophilicity and molecular weight) and *in vitro* pharmacokinetic properties (microsomal stability, solubility, plasma protein binding, and cell permeability), which is summarized in Supplementary Table S6. First, compounds **17** and **23** clearly have more lead-like properties than TH5487, including lower lipophilicity (cLogP of 3.7 and 4.0 vs. 4.6) and molecular weight (350 and 386 vs. 541 Da, respectively). In the ADME assays, the three compounds showed comparable cell permeability and moderate to good stability in human liver microsomes. TH5487 showed slightly higher stability than compounds **17** and **23**. However, the higher stability of TH5487 is likely a consequence of the very high protein binding of this compound, as compared to **17** and **23**. Notably, there were larger differences in two other important properties. In agreement with the notion that fragment-based design results in higher quality compounds, one of our inhibitors was >667-fold more soluble and showed substantially lower protein plasma binding than TH5487. These results are in agreement with previous studies of TH5487, which demonstrate that this inhibitor is poorly soluble (ref. 23) and protein plasma binding “*severely reduced its efficacy*” (ref. 24) in treatment of cancer.

It should be noted that both TH5487 and our inhibitors would benefit from further optimization. However, development of a drug candidate was out of the scope of both our and the previous study in Science by Visnes *et al.* (ref. 23). In the pursuit for the first drug targeting OGG1, access to multiple inhibitor scaffolds will be crucial and it is evident that our inhibitors have several advantages over TH5487. These new results have been added to pages 14 and 16-17 of the manuscript and are discussed on pages 24-25.

• **Application of our virtual screening technique to other targets.** One of the important findings of our work is that make-on-demand compound libraries in combination with structure-based docking calculations can accelerate fragment-based drug discovery. In the submitted manuscript, we apply our approach to OGG1, a very challenging drug target. Although we agree with the reviewer that it is “*very likely this would work for a novel target*”, we decided to further assess the potential to discover ligands of other three therapeutically relevant proteins. In the revised manuscript, we provide illustrative examples of how fragments can rapidly be elaborated for three diverse drug targets.

We first identified three proteins involved in either cancer or inflammation for which **(1)** fragment screens had been performed; and **(2)** crystal structures of protein-fragment complexes were determined. For each selected drug target (SMYD3, NUDT5, and PHIP), we assessed if make-on-demand chemical libraries could enable efficient fragment optimization. Substructure searches for the fragments were first performed, followed by docking of these compounds to the protein binding sites. Large sets of analogs were identified by these searches and many of these could be successfully docked to the binding sites with maintained overall binding mode and improved energy compared to the initial fragment (Figure 5).

These results clearly demonstrate that make-on-demand libraries are a tremendous resource for fragment elaboration. As in the case of OGG1, we also show that the efficiency of this process can be further improved by utilizing structural information to identify the relevant elaboration vectors, enabling us to rapidly retrieve the most promising candidates. As we and the reviewer expected, our approach is hence applicable to structurally diverse targets, which is summarized on pages 21-22 and in the new Figure 5. In addition to these illustrative examples, we also include guidelines for application of our approach and share computational tools to perform the calculations via our GitHub repository (<https://github.com/carlssonlab/frag2lead>, will be made available upon publication).

2. Reviewer: *“Finally the ability for this to be employed by other groups is very unlikely and this reduces impact. The authors often go from a very large number of compounds to a smaller more manageable group through their judgment and guidance of the TH5675 compound/binding site. They clearly have the expertise to do this, but providing more details on how they did this would help make the approach more applicable to others and drive impact to the field.”*

We thank the reviewer for this suggestion because we agree that it is important to enable other researchers to use our techniques. In fact, this is the reason why we primarily use molecular docking and cheminformatics tools that are free for academics, and the chemical libraries were obtained from the publicly available ZINC database. As we describe below, the steps involved in performing the fragment docking calculations and compound selection adhere to the best practices of the research field. In the revised manuscript, we have included the protocols and references needed for other researchers to perform our virtual fragment screening approach for other targets.

Compound selection from prospective docking screens are performed in two steps: **(1)** Scoring by molecular docking and **(2)** Visual inspection of top-ranked compounds. In our case, the docking calculation reduced the number of compounds to consider for experimental evaluation by 99.996% (from 14 million to only 500 molecules). In the second step of prospective virtual screens, it is common practice to visually inspect the predicted complexes and the protocols by Bender *et al.* (ref. 34) recommends ~5000 compounds (*i.e.* a ten-fold larger number than considered in our virtual fragment screen). In this step, the team of scientists performing the virtual screen attempts to compensate for the weaknesses of the docking scoring function and a published survey shows that the research field uses similar criteria (ref. 35).

In the selection step, we first focus on identifying compounds that should be excluded from further consideration and the main reasons are (ref. 34):

- **Ligand strain:** As ligand strain is not part of the docking scoring function, top-ranked compounds can have high energy conformations. This is one of the main reasons for excluding compounds from experimental evaluation.
- **Unsatisfied polar groups of the ligand:** Hydrogen bond donors or acceptors of the ligand are sometimes predicted to interact with non-polar pockets in the binding site. This likely reflects that the docking scoring function underestimates the ligand desolvation penalty. Such compounds are common and easily identified by visual inspection.
- **Binding site desolvation:** The scoring function does not take into account binding site desolvation. For this reason, non-polar groups of the ligand are sometimes positioned close to polar active site residues. This is one of the main reasons for excluding compounds after visual inspection.
- **Tautomer and ionization state of the ligand:** Compounds with multiple possible tautomerization/ionization states are analysed in detail by predicting the most common form with several programs. Top-ranked compounds that are docked in a tautomerization/ionization state that is predicted to be unlikely are deprioritized in the selection process.

After excluding compounds based on the above criteria, the overall shape complementarity of the complex is inspected and compounds that form key non-polar and polar interactions (e.g., the same hydrogen bonds as co-crystallized ligands) are favored in the selection. A few of our criteria are specific for fragment screening. For example, as fragments are only half the size of a drug, the docked compounds can occupy different subpockets of the binding site. We mainly focused on the most deeply buried subpocket of the binding site, which will be the best anchoring point of a fragment and facilitate optimization.

As we recently published a detailed description of all the above steps in Nature Protocols (ref. 34), we have added a brief summary of the selection criteria in the results section (page 7).

Minor comments:

1. Reviewer: *“The authors state, “A set of 29 compounds was selected for experimental evaluation from the 500 top-ranking clusters based on visual inspection of their complementarity to the active site.” The authors need to add more description on how they went from 500 to 29 so future researchers could implement this approach.”*

As described in the previous point, this step is common practice in the research field. We have added a brief summary of the criteria for compound selection (page 7) and we cite the guidelines by Bender *et al.*, which include more detailed protocols (ref. 34).

2. Reviewer: *“How did the authors identify the “500 top ranking clusters”?”*

We thank the reviewer for pointing out that this step was not clearly described. All docked molecules are first ranked based on their scores. In a second step, the top-ranked molecules are clustered based on Morgan2 fingerprints (1024 bits with a radius

of 2) and Tanimoto similarity, which is performed using RDKit and a toolkit from the DOCK3.7 software package: The best scoring molecule is the cluster head of the first cluster, and lower scoring molecules with a Tanimoto similarity greater than 0.5 to this molecule are included in this cluster. The first top-ranked molecule that is not part of this cluster becomes the cluster head of a new cluster and this procedure is repeated until all molecules have been assigned to a cluster. In the revised manuscript, we included a more detailed description of the clustering algorithm on page 31.

3. Reviewer: *“The xray table has several issues that make it difficult to evaluate. It appears 8CEY is reported slightly differently in regards to the way the units are presented (e.g. Rmerge, resolution). The table should have the same way of reporting the different structures. What are the outliers in 7Z5R the authors are referring to?”*

We thank the reviewer for pointing out the inconsistencies in Supplementary Table S2. The units for resolution, R_{merge} , R_{pim} and $CC_{1/2}$ are now reported in the same manner, so that the structures are directly comparable. In addition, the Ramachandran outliers for the 7Z5R structure are now correctly reported.

4. Reviewer: *“Why did the authors select the 99 and 495 μM concentrations for the DSF experiments? In the DSF experiments did any of the other 65 compounds induce a stabilization other than 1 and 2? Why did the authors only focus on these two.”*

The screening concentrations for leads (99 μM) and fragments (495 μM) were selected to correspond to conditions typical of high-throughput and fragment screening. In fragment screens, higher concentrations are typically used because the compounds bind weakly, are small (<250 MW) and have better solubility (see ref. 14). At such high concentrations, lead-like compounds (MW \approx 350 Da) can have solubility problems. As lead-like compounds also are expected to have higher potencies, lower screening concentrations are more suitable in such screens. We have included this information on page 7 in the revised manuscript.

The DSF results for all the 65 compounds are presented in Supplementary Table S1 and Supplementary Data File 1, and we have added this information on page 7. Of these, only compounds **1** and **2** displayed >1 K increase in stability. The largest shift in stability was observed for compound **1** (2.8 K) and this is the reason why we mainly focused on this compound in the optimization step, which is described on pages 10-11 of the manuscript.

5. Reviewer: *“The figure panels nomenclature is distracting in that the order from a-f is not always in one direction. Compare figure 3 vs 4.”*

We thank the reviewer for noticing these inconsistencies and we have updated the labels and legend of Figures 2 and 4.

Reviewer #2

The second reviewer appreciated our work: *“Besides the already important discovery of this compound, the paper is a valuable contribution to the growing toolbox of uHTVS methods”*, and strongly supported publication: *“I am convinced that this is a valuable methodology that is suitable for publication in Nat Comm”*. We respond to the reviewer’s questions below:

Major comments

1. Reviewer: *“Nonetheless, the authors should improve the discussion on the added value of their methodology with respect to recent workflows that apply the concept of chemical space generation via the combination of commercially available reagents, such as V-SYNTHES or SASS.”*

The strategies used in our study and the V-SYNTHES method are both based on the divide-and-conquer idea that is the foundation of fragment-based drug discovery. In the first step, both methods dock fragment-like compounds to the target binding site. The advantage of our approach is that the docked fragments are commercially available and can be tested experimentally. V-SYNTHES instead uses computationally generated fragments that will typically not be readily available. Whereas V-SYNTHES relies on the accuracy of the scoring function at this point, our approach performs experiments to identify false positives and can focus on the fragments with verified activity. In the second step of the two methods, the fragments are optimized by increasing the size of the compounds. V-SYNTHES has to rely on the binding modes predicted by docking in the optimization whereas our study utilized iteratively solved crystal structures of protein-ligand complexes. Access to multiple experimental structures was a crucial part of our optimization step because the target was very challenging with considerable induced-fit effects. Finally, V-SYNTHES is constrained to predict compounds that are available in the combinatorial library, which we found was a limitation in the optimization of the OGG1 inhibitors. The creation of tailored chemical libraries was required to obtain submicromolar OGG1 inhibitors, which has also been the case for other challenging enzyme targets (ref. 7). Based on these observations, V-SYNTHES and the approach described in this work have complementary advantages. Our strategy may be most suitable if the target is challenging (e.g. a flexible binding site) and sensitive experimental screening methods are available to measure weak binding (e.g. X-ray crystallography and NMR). The V-SYNTHES approach is preferable for proteins that are difficult to crystallize and have well-defined binding sites, which includes important target classes such as G protein-coupled receptors. In the revised manuscript, we have included a discussion that highlights differences between the methods and their strengths and weaknesses (pages 26-27).

Minor comments

1. Reviewer: *“Fig 1c: it is confusing that the authors highlight the 11B commercial space when it was not yet applied/considered at this step (initial fragment screen).”*

We agree with the reviewer and have modified Figure 1 (page 8).

2. Reviewer: *“Fig 2d: how many compounds are there in the tailored chem space? What is the number of reagents?”*

We have updated the manuscript regarding the number of reagents and products. This information is now available in Figure 2c (page 11) and Supplementary Figure S2.

3. Reviewer: *“Fig 4f: the caption erroneously specifies compound 13 instead of compound 17”*

We thank the reviewer for noticing this error and we have updated the figure legend (page 18).

Remarks on code availability

We thank the reviewer for testing our toolkit. All comments on the UniverseGenerator were addressed, and we have included a new version as part of the submission. We have successfully installed the package on three different machines, covering both Linux and MacOS operating systems. We included more detailed instructions in our repository's README file, emphasizing how to install RDKit from source and how to link the dynamic libraries. We will release our software package on the following repository (<https://github.com/carlssonlab/UniverseGenerator>, will be made available upon publication) upon publication of the manuscript. In addition, we will release an additional set of scripts that will enable others to perform cheminformatics calculations on the following GitHub repository (<https://github.com/carlssonlab/frag2lead>, will be made available upon publication)

Reviewer #3

The third reviewer was also enthusiastic about our work and described our manuscript as “*highly significant*”, “*an impressive success-story*”, and “*well-written*”. The reviewer only had minor comments to improve the manuscript, which we address below:

Minor comments

1. Reviewer: “*The retrospective virtual screening performance of the final grid selected for prospective screening should be described in more detail (for example ROC curves could be added to Supporting Information, see <https://www.ncbi.nlm.nih.gov/pmc/articles/PMC4631717/>).*”

We thank the reviewer for this suggestion. The performance of the crystal structure in molecular docking calculations was evaluated by redocking **TH5675** to the active site and assessing if the scoring function could enrich inhibitors over property-matched decoys. After optimizing sampling and scoring parameters, the docking program was both able to reproduce the binding mode of **TH5675** and identify this scaffold among decoys. In the revised manuscript, we present the virtual screening performance on pages 5-6 and include both ROC curves and results of redocking calculations in Supplementary Figure S1.

2. Reviewer: “*The virtual reactions and their SMIRKS patterns should be included in the Supporting Information.*”

We have included the virtual reactions and the corresponding SMIRKS patterns in Supplementary Figure S2 and present the results on pages 13-14.

3. Reviewer: “*Table 1, Crystal structure for THR5487: PDB 6RLW is human OGG1 and not mouse OGG1 like the other crystal structures in this work. Authors should highlight this in the Table 1. It should be noted in the text that the difference between the two proteins is very small as can be seen in Reference 23 (Figure 3A and 3B) so that reader doesn't have to dig this small piece of information from an another study.*”

We agree with the reviewer and have modified the Table 1 footnote (page 9) and noted the similarity between the human and mouse OGG1 in the results section (page 5).

4. Reviewer: “*The reason for keeping this one particular water is unclear to the reader (short additional explanation would be beneficial).*”

Analysis of binding site waters is an important part of preparation of protein structures for molecular docking calculations. This water was deeply buried and coordinated by several polar protein atoms, which suggested that it would be difficult to displace it by introducing a substituent in this pocket. For these reasons, we decided to treat the water as being part of the binding site and make it possible for docked compounds to form hydrogen bonds to the oxygen of the water. We have included a brief description of our treatment of the water in the Methods section (page 29).

5. The reviewer requested that we describe our PAINS filter: *“The authors should specify the specific version that they used (for example, RDKit version like described here https://projects.volkamerlab.org/teachopencadd/talktorials/T003_compound_unwanted_substructures.html).”*

We thank the reviewer for noticing that we hadn't specified the source of this filter. We used the PAINS filter made available by the OpenEye toolkits that is based on the work of Baell *et al.* (ref. 62). This information is now included in the Methods section (page 32).

6. The reviewer requested that we describe our clustering algorithm: *“The toolkit (RDKit?) used to do these tasks should be explicitly mentioned here.”*

We agree with the reviewer and have included a more detailed description of our clustering algorithm. The top-ranked compounds were clustered based on their docking scores and chemical similarity. The Morgan2 fingerprints used in the Tanimoto similarity calculations were calculated using RDKit and a toolkit from the DOCK3.7 software package, which is described on page 31 of the revised manuscript.

7. The reviewer requested that we describe how virtual libraries were generated: *“It is not clear where the commercially available building blocks were extracted here (from ZINC or just from Enamine?).”*

For practical reasons, we purchased building blocks from a few reliable vendors and primarily focused on Enamine in this project. The virtual libraries were generated by first downloading commercially available building blocks with suitable synthetic handles from the Enamine online database (<https://enamine.net/building-blocks/building-blocks-catalog>). In particular, we downloaded the secondary amine, arylhalide, and boronic acid subsets which are synthetically compatible with Chan-Lam, Suzuki, Ullmann, or amide couplings. In a second step, the virtual libraries were created using in-house scripts. We have also included more information regarding the generation of the virtual libraries in the supplementary information (Supplementary Figure S2), include information about where we downloaded the Enamine building blocks (page 32), and will also share the scripts required to generate the libraries via our GitHub (<https://github.com/carlssonlab/frag2lead>, will be made available upon publication).

8. Reviewer: *“Regarding references, I happened to note that there is a typo in reference #11 (the year should be 2016).”*

We thank the reviewer for noticing this error and we have updated the references accordingly.

Remarks on code availability

We thank the reviewer for testing our toolkit. All comments on the UniverseGenerator were addressed, and we have included a new version as part of the submission. We included more detailed instructions in our repository's README file and provided the necessary files to run the different code examples. The software package will be released on (<https://github.com/carlssonlab/UniverseGenerator>) upon publication of the manuscript.

Reviewer #4

The fourth reviewer wrote that one of the advantages of our proposed strategy could be *“its general application to the improvement of inhibitors of other enzymes”*. The main critical comment from the reviewer was that we need to show that our compounds have advantages over the previously identified inhibitor TH5487, which was discovered by a high-throughput screening. In the revised manuscript, we have included a comparison of our virtual screening approach and compounds to the traditional campaign and inhibitor TH5487. Our new results clearly demonstrate that our approach is more efficient and resulted in leads with improved properties compared to TH5487. We present these new results and answers to the reviewer’s other questions below:

Major comments

1. Reviewer: *“Prior investigations on the development of OGG1 inhibitors has identified a promising chemotype, with a biologically potent lead, TH5487 that has an IC₅₀ of 340 nM for the inhibition of the OGG1 glycosylase activity and has shown promising results in multiple biological end-point assays. To improve on these inhibitors, the current study utilized virtual screening and selected syntheses to test predicted efficacy for potential leads. However, enthusiasm for these investigations were greatly tempered by the findings that at the end of the experimental pipeline, the best, newly described molecules had IC₅₀s less potent (600 nM) than the previously described inhibitor.”*

We thank the reviewer for these comments as they prompted us to further analyze the quality of our inhibitors and those previously identified. As is common in many research papers in the field, we primarily described optimization of compound potency in our manuscript and, admittedly, overlooked reporting other important properties. Hit-to-lead optimization is a complex process and it is now widely accepted that physicochemical (e.g. lipophilicity and molecular weight) and pharmacokinetic properties (e.g. solubility and metabolic stability) are equally important as potency. In fact, the disproportionate focus on inhibitor potency in the pharmaceutical industry is considered a significant factor contributing to the high attrition rates in drug discovery. As described in Hann’s seminal paper (<https://doi.org/10.1039/C1MD00017A>, ref. 38), there is a *“tendency to build potency into molecules by the inappropriate use of lipophilicity which leads to the premature demise of drug candidates.”* In the last paragraph of Hann’s paper, he concludes:

“Starting with the smallest possible lead (i.e. fragments) and striving to maintain their fitness through the use of various indices is now accepted as a key approach in a more holistic approach to contemporary drug discovery. The absolute need for potency should not be as dominant an attractor as we often allow it to become at the expense of other characteristics of a good drug.”

Interestingly, Hann (ref. 38), as well as other papers (ref. 14), therefore suggests that using a fragment-based approach instead of traditional screening can lead to compounds of higher quality. As our inhibitors and TH5487 have comparable potencies (TH5487 has 2-fold better IC₅₀ value, our compounds have 2-fold better thermal stabilization than TH5487 in the DSF assay), we instead compared other important physicochemical and *in vitro* pharmacokinetic properties.

According to Hann's analysis of how to increase success-rates in drug discovery, molecular weight (MW), lipophilicity (LogP), and Lipophilic Ligand Efficiency (LLE) are the key properties to control in hit-to-lead optimization. Encouragingly, compounds **17** and **23** have more lead-like properties than TH5487: Lower cLogP (3.7 and 4.0 vs. 4.6), MW (350.4 and 386.4 vs. 541.2 Da, respectively), and higher LLE (2.5 and 2.2 vs. 1.9). In addition, our compounds also showed improved properties over TH5487 in the *in vitro* pharmacokinetic assays. Our compound **17** is >667-fold more soluble and shows substantially lower protein plasma binding than TH5487, a limitation of this inhibitor that "severely reduced its efficacy" (ref. 24) in treatment of cancer. To summarize, our novel scaffold has advantages over the previously discovered inhibitor, which can be attributed to our fragment-based design approach. Our novel inhibitor scaffolds can accelerate the development of the first drug targeting OGG1. Perhaps even more importantly, our technique can be applied to numerous other drug targets, which we illustrate for three other proteins. We have introduced the new data in the results (pages 14 and 16-17) and discussion (pages 24-25) of the revised manuscript.

Minor comments

1. Reviewer: *"Since the biological studies involving OGG1 inhibitors have almost exclusively focused on TH5487, it is not clear why TH5675 is used as the gold-standard from which to simulate fragment screening. It would seem much more germane to use structural information concerning this complex than that currently chosen. Given that the starting structure for the docking screens was that of 6G3Y, chain A, and the authors acknowledge that analyses of OGG1 structures reveal that "the active site is highly polar and adopts different shapes in the complexes with the inhibitor and DNA", in the Results section, the authors should provide greater justification for the use of this as their starting structure. Again, it may suggest that the structure of the biologically-relevant inhibitor would be a far-preferable starting structure."*

We completely understand that the reviewer found our choice of OGG1 structure to be odd, but there is a very simple explanation: At the time of the virtual screen (Dec 2018), the crystal structure of OGG1 in complex with TH5487 was not available. In fact, the mOGG1-TH5675 complex was the only available structure with a small molecule bound, and this was the reason why this structure was used in the first virtual screen. We have updated the results with this information (page 5).

Retrospectively, we analysed if using the crystal structure of human OGG1 bound to TH5487 would have led to a better result in the docking screen. An alignment of the mouse (PDB ID: 6G3Y) and human (PDB ID: 6RLW) OGG1 structures showed that these enzymes have close to identical active sites bound to TH5487 and TH5675 (RMSD = 0.5 Å, C α atoms). This can be explained by the fact that TH5487 and TH5675 are congeneric chemical structures. In summary, using the crystal structure with TH5487 bound would likely not have influenced the success-rate of the docking screen. We have noted the high similarity between mouse and human OGG1 on page 5.

2. Reviewer: “Comparisons of the IC₅₀ data with the ΔT_m values, does not reveal any predictive trends; could the authors speculate on the seemingly lack of correlation of those values? It would seem reasonable that protein stabilization and catalytic inhibition would have a highly significant correlation.”

This is an interesting question and we address the reviewer’s two comments below:

• **Is there a correlation between the IC₅₀ data with the ΔT_m values?**

As we had not analysed the relation between our DSF and pIC₅₀ values, we first investigated if the data “does not reveal any predictive trends”, which was the reviewer’s first comment. In this analysis, we used all the compounds from our series with determined IC₅₀ and ΔT_m values (15 compounds). We found that there is a good correlation between pIC₅₀ and ΔT_m values for our inhibitor series ($R^2 = 0.69$), which is shown below:

In the figure above, we have also included TH5487 (blue point), which belongs to a different chemical scaffold. Interestingly, this data point appears to be an outlier in our analysis (excluded from correlation coefficient). Despite being potent (IC₅₀ = 340 nM, ΔT_m = 4.3 K), this inhibitor does not stabilize OGG1 as much as our inhibitors (e.g., compound **23** with IC₅₀ = 600 nM and ΔT_m = 9.0 K). We read relevant literature to understand the origin of its observation, which is discussed in the following section.

• **Should we expect inhibition and protein stabilization to be correlated?**

In the second comment, the reviewer writes that “it would seem reasonable that protein stabilization and catalytic inhibition would have a highly significant correlation.” We investigated this question in more detail and found the reviewer’s point of view does not align with results in the scientific literature. We could find several articles in which binding affinity or inhibition did not correlate with thermal shifts, and a review by Gao *et al.* on the theory of DSF summarizes these observations in the conclusion section (<https://link.springer.com/article/10.1007/s12551-020-00619-2>, ref. 36):

“In terms of ligand-binding validation, although many successful cases have been reported in the literature, it is still important to be aware that this correlation typically occurs for similarly structured compounds within a series”

Our observation that pIC_{50} and ΔT_m values correlate for our series, but not TH5487 as we noted above, is thus in agreement with scientific literature and expected based on the limitations of DSF. Several papers have discussed the origin of this effect, but this is out of the scope of our study. One interesting hypothesis is that the thermodynamic profiles of the inhibitors, *i.e.* the enthalpy-entropy balance, differ (see <https://doi.org/10.1177/10870571111399573>). Compounds with different thermodynamic profiles can be useful in drug discovery, which again demonstrates that TH5487 and our compounds have complementary advantages. Another contributing factor to TH5487 being an outlier could also be its poor solubility compared to our fragment series, which may become limiting at the concentration at which the DSF analyses were performed. Thus, our fragment-based approach yields ligands which can be validated in orthogonal assays without being limited by poor physicochemical properties.

In the revised manuscript, we have noted that the difference in ΔT_m for our inhibitors and TH5487 may be due to that the compounds belong to different chemical series (page 14).

3. Reviewer: *“Comparison of data presented in Tables 1 and 2, it is unclear why the authors report IC₅₀ values in Table 1 and pIC₅₀ values in Table 2. The text associated with Table 2 presents the IC₅₀, not the pIC₅₀ determinations. Unless there is a compelling reason to change to pIC₅₀, it would be better to consistently report the IC₅₀ data.”*

We thank the reviewer for noticing the inconsistency in how we presented inhibitory potencies. After discussing this question with the assay team, we found that there is no clear consensus on whether to use IC_{50} or pIC_{50} values. On one hand, there are formal reasons to prefer pIC_{50} values, such as curve fitting being based on the log scale and the more accurate representation of standard error. On the other, many scientists find it more intuitive to discuss concentrations (IC_{50} values) rather than using a logarithmic scale (pIC_{50}). Ultimately, this seems to be a matter of preference. To ensure consistency in our manuscript, we have reformatted all tables to report pIC_{50} values, while IC_{50} values are used in the manuscript text.

4. Reviewer: *“In the text, the authors reference fold improvements over that of their reference compd 1- given that the IC₅₀ of compd 1 was reported as >99 μM , how can fold improvements be calculated given that no definitive IC₅₀ was determined? It would seem much more reasonable to assess the success of the screenings based on the previously reported IC₅₀ for TH5487 being 340 nM, with a change in $T_m = 4.3$.”*

We thank the reviewer for pointing out that we should clarify how we calculated the potency improvement. The inhibitory potency of compound **1** was too weak to reliably determine an IC_{50} value. This is not unusual in fragment screening because of the small size of the compounds. However, since the percent inhibition is less than 50% at 99 μM , we can be certain that the IC_{50} is greater than 99 μM ($IC_{50} > 99 \mu M$). As a result, we cannot report the exact fold improvement in potency compared to our most potent inhibitor ($IC_{50} = 0.6 \mu M$). The improvement achieved in our fragment-to-lead optimization equals the ratio of the inhibitory potencies. If we knew the exact potency of compound **1**, we would also report an exact fold-improvement. For example, if the

IC₅₀ value of compound **1** would be 150 μM, the fold-improvement would be 250. If the IC₅₀ value was twice as high (200 μM), the improvement would be 500-fold. In other words, >165 is a conservative and the best estimate we can provide of the improvement.

Finally, we would like to point out: **(1)** that it is common practice in the field of fragment-based discovery to report improvements in potency compared to the starting point, and **(2)** that many other studies report “larger than” (“>”) improvements of potencies. For example, a yearly survey of fragment-based drug discovery in the *Journal of Medicinal Chemistry* reports the “activity fold change” in Table 2 for 28 case studies (<https://pubs.acs.org/doi/10.1021/acs.jmedchem.2c01827>). The authors **(1)** report the fold-change compared to the starting-point and; **(2)** they, as in our paper, use greater than (“>”) in relevant cases (e.g., case 8 in Table 1). The fold improvement compared to the starting point is an important metric and >100-fold is considered to be a significant improvement by this review article. For these reasons, we maintained the comparison to the fragment and report >165-fold improvement, in agreement with previous work. As discussed above (point 1), we of course also agree with the reviewer that a comparison to previously discovered inhibitors is also important, and we include an extensive comparison to TH5487 in the revised manuscript, which is presented on pages 14 and 16-17.

UPPSALA
UNIVERSITET

SciLifeLab

December 6th, 2024

We look forward to publishing our paper in Nature Communications ("*Virtual Fragment Screening for DNA Repair Inhibitors in Vast Chemical Space*", Manuscript: NCOMMS-24-23933). We have addressed the new comments regarding code availability. Please find below point-by-point responses to the referees with their comments in **Blue**.

Reviewer #3

The reviewer had additional suggestions on further improving the code availability: "*I was able to compile the code from reviewer ZIP-file and run the demo example, but noticed couple of typos/bugs which are good to fix before the source code is released to the public*"

Remarks on code availability

We thank the reviewer for spotting these typos and have implemented the suggested changes in our new version of the repository's README file.